# Tumor-Associated Macrophages: Polarization, Immunoregulation, and Immunotherapy

**DOI:** 10.3390/cells14100741

**Published:** 2025-05-19

**Authors:** Abdullah Farhan Saeed

**Affiliations:** Department of Internal Medicine, University of Michigan, Ann Arbor, MI 48109, USA; abdullfs@umich.edu

**Keywords:** macrophage, polarization, TAM, immune regulation, immune checkpoint, cancer, immunotherapy

## Abstract

Tumor-associated macrophages’ (TAMs) origin, polarization, and dynamic interaction in the tumor microenvironment (TME) influence cancer development. They are essential for homeostasis, monitoring, and immune protection. Cells from bone marrow or embryonic progenitors dynamically polarize into pro- or anti-tumor M2 or M1 phenotypes based on cytokines and metabolic signals. Recent advances in TAM heterogeneity, polarization, characterization, immunological responses, and therapy are described here. The manuscript details TAM functions and their role in resistance to PD-1/PD-L1 blockade. Similarly, TAM-targeted approaches, such as CSF-1R inhibition or PI3Kγ-driven reprogramming, are discussed to address anti-tumor immunity suppression. Furthermore, innovative biomarkers and combination therapy may enhance TAM-centric cancer therapies. It also stresses the relevance of this distinct immune cell in human health and disease, which could impact future research and therapies.

## 1. Introduction

Cancer ranks high among global public health concerns and is leading to high fatalities in multiple countries. In 2025, 2,041,910 new cancer cases and 618,120 cancer-related deaths are projected in the US [1]. The high cost of treating cancer patients and their families, as well as the whole economy, has been a significant drag on healthcare investment. Accordingly, to further decrease mortality and morbidity rates, tumor prevention and treatment are of utmost urgency. Despite the widespread use of chemotherapy, radiation therapy, and surgery as therapies for cancer, some patients have had unsuccessful outcomes [2,3].

In animals, macrophages penetrate various tissues, where they co-develop with the organs they occupy. Until their environment changes, these macrophages generally keep their numbers and morphologies the same. There are three main types of macrophages found in almost every tissue: those that originate in the blood or bone marrow, those that reside in the tissue itself, and those that arise in the bone marrow but do not produce any inhibitory cells [4]. Due to their phenotypic and functional diversity, macrophages are essential to innate and adaptive immunity. Embryonic hematopoietic stem cells become lung alveolar macrophages, bone osteoclasts, and liver and Kupffer cells. Postnatal bone marrow precursors, specifically monocytes, may become macrophages in tissue inflammation or a steady state [5].

Immunology, tissue and systemic inflammation, and regeneration are all processes in which macrophages participate. They perform phagocytosis, antigen presentation, microbial cytotoxicity defense, cytokine and complement components, and other tasks [6]. Likewise, immunological tolerance, tissue repair, and inflammation suppression are all achieved by macrophages [7]. Moreover, TAMs help establish TME. Many cancers include TAMs that increase tumor development, invasion, metastasis, and treatment resistance [8]. Furthermore, macrophages’ functional differences are linked to plasticity, and TME chemicals modulate their functional phenotype [9].

Additional therapeutic methods to develop after conventional therapy are chimeric antigen receptor (CAR)-T cell therapy and immune checkpoint inhibitors. Some advanced-stage tumors have demonstrated encouraging results when treated with immunotherapies, such as CAR-T cell therapy and programmed death receptor 1 (PD-1) inhibitors, which have recently gained clinical approval [10,11,12,13]. Still, there is room for improvement in the effectiveness of PD-1 inhibitors, as they were only partially effective in a subset of cancer patients [14]. Similarly, TAMs are pivotal TME regulators, exhibiting remarkable heterogeneity in origin, polarization states, and functional roles, influencing cancer progression and therapeutic outcomes [15]. Derived primarily from circulating monocytes or embryonic precursors, TAMs adopt context-dependent phenotypes, ranging from pro-inflammatory (M1-like) to immunosuppressive (M2-like)-shaped, through dynamic interactions with tumor cells, stromal components, and metabolic regulation [16,17]. Recent advances in single-cell profiling and spatial transcriptomics have refined TAM characterization, revealing subsets with distinct prognostic and therapeutic implications [15]. Within the TME, TAMs facilitate tumor growth by promoting angiogenesis, metastasis, and immune evasion through direct crosstalk with malignant cells and suppressing cytotoxic T-cell activity [17]. Meanwhile, immunotherapies targeting PD-1 and programmed cell death ligand 1 (PD-L1) checkpoint pathways and TAM-modulating strategies show promise. Debates persist over the efficacy of TAM depletion versus functional reprogramming, underscoring the need for precision approaches tailored to TAM plasticity and tumor-specific contexts [18,19].

Investigating the TME to increase the response rate and create novel cancer immunotherapy using recent advances is essential. Research indicates that macrophages are among the most important immune cells found in the TME [16]. Macrophages initially clear tumor cells through phagocytosis. However, exposure to factors in the TME induces their transformation into M2-polarized TAMs, which promote metastasis and tumor growth by suppressing immunity, inducing angiogenesis, and supporting cancer stem cells [20]. Studying macrophages in the TME is crucial, and future anti-tumor efforts may benefit from targeting macrophages. This review focuses on TAM polarization, characterization, immunoregulation, and crosstalk with other immune cells in the TME. Moreover, the article describes recent technologies enabling precision therapeutics, single-cell analysis, and spatial transcriptomics. These are vital to dissecting TAMs’ heterogeneity and enhanced characterization, flexibility, and interactions in the TME.

### Search Strategy

For searching the literature for our review, we considered authenticated databases, such as Google Scholar and PubMed. We used the string “TUMOR ASSOCIATED MACROPHAGE”, “POLARIZATION”, “IMMUNOREGULATION”, AND “IMMUNOTHERAPY IN CANCER”. We excluded duplicate records, reviews, book chapters, and titles that were considered irrelevant. Hence, only relevant papers were specifically chosen for inclusion in our review due to their focus on TAMs, immunoregulation, and immunotherapy. This concise approach ensures that the literature synthesized in our review is substantial and directly relevant to TAMs and immunoregulation in the context of cancer therapy.

## 2. Macrophage Origin, Polarization, and Characterization

To fully appreciate the diverse roles of macrophages in health and disease, it is essential to first examine their developmental origins, the approaches underlying their polarization, and the mechanisms for their novel characterization. Monocytes in the bloodstream are the progenitors of macrophages, and there is a great deal of diversity among these immune cells [21]. They may originate from tissue macrophages or blood monocytes drawn to chemokines like CCL2 or CSF-1 [22]. Classically activated macrophages (M1) and alternatively activated macrophages (M2) are two types whose phenotype and function could be identified [23].

### 2.1. Macrophage Differentiation in the TME

Environmental signals cause macrophages to differentiate into distinct phenotypes for host defense, wound healing, immunological control, and cancer [24]. The malleability of undifferentiated macrophages (M0) allows them to polarize into two types of activation: conventional (M1) and alternative (M2) [25]. IFN-γ and LPS from bacteria activate M1 macrophages, which release inflammatory molecules, such as IL-1β, IL-6, and TNF-α. They have pro-inflammatory and anti-tumor characteristics. When tumors grow, spread, and evade the immune system, IL-4 and IL-13 released by T-helper 2 (Th2) cells influence M2 macrophages. In addition, they release anti-inflammatory components, such as IL-10 and TGF-β [26].

Furthermore, abnormalities in the *IKKβ/NF-κB* pathway may cause TAM polarization [27]. IL-4 activates its receptor to phosphorylate *STAT6*, driving M2 macrophage polarization through the *JAK/STAT6* pathway. To induce polarization, phosphorylated *STAT6* binds to *KLF4* and *PPAR-γ* simultaneously [28]. M2-like macrophages become M1-like by blocking IL-4’s receptor [29]. Multiple mediators, including IL-4, TGF-β, IL-10, and *BMP-7*, activate M2 polarization via the *PI3K/Akt* pathway [30]. Non-coding RNA (lncRNA)-Xist knockdown or miR-101 overexpression in M1-like macrophages suppresses *C/EBPα* and *KLF6* synthesis, leading to M1 to M2 polarization [31]. Anti-tumor therapies that regulate M1/M2 polarization are promising owing to TAMs’ higher adaptability.

It’s important to note that macrophages are not merely M1 and M2 immune cells. IL-4/IL-13, LPS/IL-1 receptor, and IL-10 promote activated M2 macrophages into M2a, M2b, and M2c, respectively [32]. According to Mantovani et al. [33], M2a and M2b macrophages modulate immunological response and enhance the response of Th2 cells. Meanwhile, M2c suppresses the immune response and remodels tissue. Moreover, Toll-like receptors activate M2d macrophages (TAMs) expressing VEGF and IL-10 [34]. In addition, M2d macrophages promote tumor growth and angiogenesis [35]. Figure 1 depicts inducers, surface indicators, and cytokines. Targeting TAMs in the TME is becoming more critical, as macrophage activation may control tumor growth and inflammation.

### 2.2. Recent Technological Advances in TAM Characterization

Recent advancements in TAM characterization leverage integrated multi-omics approaches and spatial technologies to uncover novel biomarkers and functional heterogeneity. These innovations are refining our understanding of TAM subsets and their clinical implications [36].

#### 2.2.1. Integrated Multi-Omics Strategies

Combining mass cytometry (CyTOF) and single-cell RNA sequencing (scRNA-seq) enables simultaneous protein and gene expression profiling at the single-cell level. They reveal TAM subpopulations with distinct phenotypic and transcriptomic features [37]. This approach is applied to study lung adenocarcinoma and pancreatic ductal adenocarcinoma [37]. Techniques like spatial barcoding and AI-driven image analysis (e.g., HALO platform) map TAM distribution within TMEs, linking spatial patterns to functional states. For example, spatial transcriptomics in glioblastoma (GBM) identified M2 TAM-associated genes correlated with poor prognosis [38].

Similarly, spatial omics, such as imaging mass cytometry (IMC) and scRNA-seq, revealed macrophage heterogeneity and their immunosuppressive roles in breast cancer. It linked high macrophage infiltration to recurrence risk and altered survival outcomes [39]. In hepatocellular carcinoma (HCC), single-cell transcriptomics integrated with Mendelian randomization identified SPP1^+^ TAMs as key mediators of immune evasion. At the same time, trajectory analysis mapped their differentiation from monocytes to proliferative subtypes, offering insights into TAM plasticity [40]. Similarly, multi-omics profiling in HCC uncovered GRN^+^ macrophages as drivers of immunosuppression, providing potential therapeutic targets. For colorectal cancer, combining scRNA-seq and bulk RNA-seq enabled the development of a prognostic signature (TAMM2RS) based on M2-TAM biomarkers like *DAPK1* and *TRAF1*, which correlate with patient survival [41]. Pan-cancer single-cell analyses further identified TAM clusters linked to immunotherapy response, with pro-inflammatory subsets associated with poorer survival [42]. These studies underscore how multi-layered omics approaches enhance our understanding of TAM biology, clinical stratification, and therapeutic opportunities.

#### 2.2.2. AI-Driven Image Analysis

Multiplex immunohistochemistry (mIHC) and automated platforms (e.g., HALO) use artificial intelligence (AI) for tissue segmentation and quantitative analysis of TAM density, cluster morphology, and stromal interactions [43]. Likewise, recent advancements in AI-driven image analysis have significantly enhanced the characterization of TAMs, particularly when predicting their spatial distribution, phenotypic states, and clinical implications. A recent study developed a Masked Autoencoder (MAE)-ResNet model to predict M2 macrophage levels in HCC using whole-slide histopathological images. It achieved an AUC of 0.73, linking higher M2 abundance to poorer prognosis [44]. Another approach integrated convolutional neural networks (CNNs) and generative adversarial networks (GANs) to infer spatial transcriptomic features from H&E-stained images. The work enabled cost-effective analysis of TAM heterogeneity and its role in tumor ecosystems [45]. Also, deep learning frameworks like Mask R-CNN and commercial tools (e.g., inForm software v3.1.0) automated TAM segmentation and classification in mIHC. The study revealed spatial patterns correlated with survival outcomes [45]. Additionally, novel imaging tracers targeting macrophage biomarkers (e.g., CD163) combined with AI analysis enabled dynamic monitoring of TAM localization during immunotherapy. This investigation offered insights into treatment resistance [46]. These methods highlight AI’s growing role in bridging histopathology with multi-omics data to refine prognostic and therapeutic strategies.

#### 2.2.3. Functional Enrichment and Survival Analysis

Recent studies highlight using integrated computational frameworks like MetaTiME for functional enrichment analysis and machine learning models. The models include Random Forest (Macro.RF) for survival analysis, alongside scRNA-seq and bulk RNA-seq data to characterize TAM heterogeneity and prognostic impact [47,48,49]. Moreover, tools like DAVID and LASSO regression analyze differentially expressed genes/proteins in TAM clusters, associating specific subsets with immunosuppressive pathways or survival outcomes [50]. Additionally, studies identify distinct TAM subpopulations linked to survival outcomes, such as pro-inflammatory TAM clusters associated with shorter overall survival (OS) in pan-cancer analyses [42]. Correspondingly, LUAD-specific M2-like TAM subsets correlate with poor prognosis [51]. Functional enrichment techniques like KEGG pathway analysis and Gene Set Variation Analysis (GSVA) reveal TAM involvement in metastasis-related pathways (e.g., SPP1 signaling) [51]. Novel prognostic models, including 10-gene TAM risk signatures, demonstrate robust predictive power for survival and immunotherapy response stratification [51]. These approaches integrate scRNA-seq-derived TAM biomarkers with clinical data, enabling precise patient risk categorization and therapeutic targeting [48,51,52].

#### 2.2.4. Prognostic Biomarkers in Hematologic Malignancies

Elevated levels of sCD163, CCL2, and CCL4 predict shorter time to treatment (TTT) and OS in Waldenström’s macroglobulinemia (WM), reflecting TAM-mediated chemotaxis and immune evasion [53]. In glioblastoma, scRNA-seq and bulk RNA-seq integration identified 16 M2 TAM-related genes (e.g., SPP1, *C5AR1*) linked to chemotherapy resistance and poor prognosis [54]. In colorectal cancer (CRC), SPP1, *C5AR1*, *MMP3*, *TIMP1*, and *ADAM8* were validated as macrophage-related biomarkers influencing tumor progression [55].

Similarly, a new study integrated scRNA-seq with bulk transcriptomic data to define six TAM subsets in lung adenocarcinoma, culminating in a 10-gene risk signature that predicts survival and immunotherapy response [51]. Pan-cancer analyses using scRNA-seq further revealed conserved pro-inflammatory and pro-tumor TAM clusters across nine cancer types. Spatial profiling links these clusters to distinct TMEs and therapeutic sensitivities [51]. Additionally, multi-omics approaches combining weighted gene co-expression networks (WGCNA) with experimental validation identified biomarkers like *ALOX5* (associated with invasion) and *ADH1A* (linked to chemosensitivity) in pancreatic cancer. These are further validated using RT-qPCR [56]. Spatial technologies also highlighted SPP1^+^ macrophages as markers of poor prognosis, while cytometry-based methods (e.g., imaging mass cytometry) enabled high-resolution TAM phenotyping [57]. These advances underscore the shift toward precision biomarkers that refine prognosis and guide immunotherapy strategies.

#### 2.2.5. Pan-Cancer Biomarkers

In hypoxia- and angiogenesis-related genes, signatures like ARSig (angiogenesis-related genes) in soft-tissue sarcomas and ICD (immunogenic cell death) subtypes in renal cancer predict immunotherapy response [58]. In inflammation-associated genes, *C15orf48* in thyroid cancer and *DARS2* in bladder cancer are emerging prognostic markers tied to TAM activity [58].

A recent pan-cancer scRNA-seq study across nine cancer types identified two conserved TAM clusters. Pro-inflammatory (*TNFSF10*+) and pro-tumor (SPP1^+^) subtypes were linked to distinct TME profiles, immunotherapy responses, and clinical outcomes [42]. These findings were validated in bulk RNA-seq data from 9164 TCGA tumors. The study enabled the stratification of 32 cancer types based on TAM-driven molecular subtypes [42]. Parallel proteogenomic analyses of 1056 tumors across 10 cancers revealed seven pan-cancer immune subtypes, with phosphoproteomic profiling uncovering kinase activity patterns tied to TAM-mediated immune evasion [59]. Additionally, spatial transcriptomics and machine learning approaches enhanced the resolution of TAM spatial organization and its correlation with digital pathology features. These multi-omics frameworks provide actionable biomarkers for predicting immunotherapy efficacy and designing macrophage-targeted therapies [59].

#### 2.2.6. Clinical Implications of Technological Advances

These technologies and biomarkers enable precise TAM subpopulation targeting. For example, blocking SPP1 or *CCL2*/*CCL4* pathways could disrupt TAM recruitment in CRC or WM [53]. Meanwhile, spatial transcriptomics-guided therapies may improve tumor outcomes by countering M2 TAM-driven resistance [47]. Ongoing challenges include standardizing cross-method validation and translating findings into therapies that exploit TAM heterogeneity.

Furthermore, these approaches enable high-resolution profiling of TAM heterogeneity, revealing distinct subsets with unique molecular signatures linked to tumor progression and patient prognosis [60]. For example, scRNA-seq has identified TAM subpopulations associated with aggressive cancer subtypes. This facilitates the development of prognostic biomarkers, such as CD206/CD68 ratios in colon cancer [61]. Clinically, these technologies inform precision interventions by uncovering therapeutic targets like the CD47/SIRPα axis and CCL2/CCR2 signaling. The targets are explored in over 700 trials to reprogram TAMs into anti-tumor phenotypes or block their recruitment [52,60]. Additionally, IMC provides spatial insights into TAM interactions within the TME, guiding combination therapies with immune checkpoint inhibitors [62]. These innovations enhance patient stratification and pave the way for personalized immunotherapies that leverage TAM plasticity to improve treatment responses [60,61].

## 3. Macrophages Inside of the TME Accelerate Tumor Development

A comprehensive understanding of macrophages’ origin, polarization, and characterization sets the stage for examining how these highly plastic immune cells actively contribute to tumor progression. Malignancy recruits and reprograms within the TME [63]. At various points during tumor formation, macrophages play an essential role. Through the first steps of tumor growth, immune cells, such as macrophages, migrate into the tumor stroma in response to cytokines and exosomes released by tumor cells. After they arrive, circulating monocytes recruit to the TME via chemokines like CCL2 and CSF1 [63]. They aid migration, metastasis, and tumor growth [35]. Moreover, macrophages can induce tumor necrosis through assertive phagocytosis [64]. Nevertheless, there is evidence that TAMs play a substantial role in further tumor advancement. In cancer, TAM increases cancer cell expansion and proliferation, angiogenesis, and lymphangiogenesis and suppresses the effector T cell immune response [65].

At the outset of lung cancer, TAM exhibits characteristics that are thought to be pro-inflammatory and anti-tumor (M1 type). Still, as the disease advances, TAM begins to show characteristics thought to be pro-inflammatory and tumor-promoting [20]. TAM can potentially regulate the immune system and non-immune processes to promote tumor formation [66,67,68]. One example is TAM secretion of many pro-angiogenic factors, which aid in tumor angiogenesis and metastasis [69].

Half of the TME’s cells are macrophages, mostly M2 phenotypes. A negative correlation has been observed between tumor micro-vessels, the number of macrophages in the TME, and survival results in patients with non-small cell lung cancer (NSCLC) [70,71]. Recently, an explosion of research saw the use of next-gen and single-cell sequencing technologies. These unveiled the TAM’s complex management of the ever-changing cancer environment [23,71]. The primary objective of this section is to deliver an overview of TAM and its role in tumors.

### 3.1. Implications of M1-Type TAM for the Regression of Tumors

TAMs’ primary deleterious process is suppressing anti-tumor immunity. One immunostimulatory cytokine that TAM may be able to lower is IL-12 production. This cytokine can stimulate NK and cytotoxic CD4^+^ T cells, potentially eradicating tumors [72]. In addition, TAMs’ production of immunosuppressive elements, including IL-10, transforming growth factor-β, and prostaglandin E2, might have a role in cancer formation [20,73,74].

Arginase 1 (ARG1) is a hydrolase that regulates the breakdown of L-arginine; it is one of the enzymes that TAM may directly block for T cell activity via IL-4 and IL-10. Likewise, hypoxia is the mediator of many signaling pathways that generate ARG1. This impacts T cell function by reducing the activity of the semi-essential amino acid L-arginine [74]. Additionally, by blocking the expression of PD-L1 and B7 homolog 1 on T cells, TAM may enhance T cell apoptosis [23,74].

### 3.2. M2-Type TAM’s Role in Driving Tumor Growth

Pro-inflammatory cytokines, such as TNF-α, IL-6, and IL-11, mediate the tumor-promoting function of M2 macrophages in cancer cells by activating the *NF-κB*/*STAT3* pathways [20,23,67,74,75]. Furthermore, M2 TAM accelerated tumor growth via elevated VEGF-A and VEGF-C expression, which enhanced angiogenesis and lymphangiogenesis [67,69,74].

### 3.3. Dual Roles of TAMs in Pro-/Anti-Tumor Immunity

TAMs exhibit a dual role in tumor immunity, balancing pro-tumor and anti-tumor functions depending on their polarization state and interactions within the TME. M1-polarized TAMs act as tumor suppressors through direct cytotoxicity, killing tumor cells via reactive oxygen/nitrogen species and TNF-related apoptosis-inducing ligand (TRAIL) [18]. Similarly, in activating adaptive immunity, antigens are presented to T cells via MHC molecules. Secreting pro-inflammatory cytokines (e.g., IL-12, IFN-γ) enhances CD8^+^ T cell and NK cell activity [76]. Counteracting regulatory T cells (Tregs) and inhibiting angiogenesis in early tumor stages disrupt immunosuppression [77].

M2-polarized TAMs drive tumor progression through immunosuppression by secreting IL-10 and TGF-β to inhibit CD8^+^ T cells and recruit Tregs via CCL17/CCL22 [78]. Upregulating PD-L1 and CTLA-4 ligands suppresses T cell function through checkpoint inhibition [79]. Depleting L-arginine via arginase 1 blocks CD8^+^ T cell proliferation [80]. TME remodeling promotes angiogenesis using VEGF, EGF, and metastasis through matrix metalloproteinases (MMPs) [81]. They enhance tumor cell resistance to therapy by activating the *PI3K/Akt* and *JAK/STAT* pathways [79]. Metabolic reprogramming induces autophagy in cancer cells to foster drug resistance [82]. Crosstalk with tumor cells through exosomes and soluble factors (e.g., CCL2, CSF-1) reprograms TAMs toward pro-tumor phenotypes [83].

### 3.4. TAM Regulation in Solid and Liquid Tumors

TAMs are critical regulators of tumor progression, with distinct roles in solid versus liquid tumors. While most research focuses on solid tumors, emerging evidence suggests that TAMs also influence hematologic malignancies [84].

#### 3.4.1. TAMs in Solid Tumors

TAMs infiltrate the stromal and hypoxic regions of solid tumor masses, interacting with cancer-associated fibroblasts (CAFs), endothelial cells (ECs), and extracellular matrix components [78]. They are often polarized to an M2-like immunosuppressive phenotype, promoting angiogenesis through VEGF, MMPs, and hypoxia-inducible factor-1α (HIF-1α) [82], metastasis, and immune evasion [78]. Perivascular TAMs (PvTAMs) near blood vessels drive angiogenesis and chemotherapy resistance [78]. Furthermore, they facilitate tumor cell intravasation via TMEM structures (TME of metastasis) and promote pre-metastatic niche formation [82]. Likewise, TAMs produce IL-10, TGF-β, and checkpoint inhibitors (e.g., PD-L1), suppressing cytotoxic T cell activity [85]. Therapeutic strategies focus on depleting TAMs, blocking recruitment (e.g., CCR2/CSF1R inhibition), or repolarizing M2-like TAMs to anti-tumor M1 phenotypes [82]. Similarly, anti-angiogenic therapies targeting TAM-derived VEGF have been explored [86]. High TAM infiltration correlates with poor prognosis, advanced stage, and therapy resistance in cancers like breast, prostate, and colorectal cancer [87].

#### 3.4.2. TAMs in Liquid Tumors

In hematologic malignancies (e.g., leukemia, lymphoma), TAMs reside in bone marrow, blood, or lymphatic systems. Their interactions are less well-characterized. Nevertheless, it may involve supporting cancer cell survival in niches or modulating immune responses in fluid microenvironments [88]. Research suggests TAMs may contribute to immune evasion and drug resistance in leukemia/lymphoma by altering bone marrow stromal interactions or secreting survival factors for malignant cells [88,89]. Their role in angiogenesis is less relevant due to the fluid nature of these tumors. Potential targets include bone marrow niche interactions or TAM-mediated survival pathways. For example, blocking cytokine loops (e.g., IL-6/JAK/STAT) might disrupt TAM support for leukemic cells [90]. TAM density and phenotype associations are less clear, although some studies suggest immunosuppressive macrophages may worsen outcomes in lymphomas [91].

Nonetheless, TAMs in solid tumors have been well-studied as drivers of angiogenesis, metastasis, and immunosuppression. Their roles in liquid tumors remain under-researched. Differences in TME architecture and functional demands (e.g., angiogenesis vs. fluid dissemination) likely shape these disparities and require further investigation.

### 3.5. Regulation of TAM-Associated Metabolic Reprogramming in TME

TAMs prioritize aerobic glycolysis (the Warburg effect) even under normoxic conditions, marked by upregulated enzymes like hexokinase-2 (HK2), enolase 1 (ENO1), and pyruvate kinase M2 (PKM2) [92]. This glycolytic flux generates lactate, stabilizing HIF-1α and mTOR pathways to reinforce pro-tumoral polarization [93]. Moreover, it competes with cancer cells for glucose, driving TAMs toward oxidative phosphorylation (OXPHOS) in glucose-limited niches [94].

In TAM-associated amino acid metabolic regulation, glutamine supports M2-like polarization via tricarboxylic acid (TCA) cycle intermediates and fatty acid oxidation (FAO). In contrast, L-methionine sulfoximine (MSO) regulates TAMs’ glutamine metabolism and orchestrates them to polarize toward the M1 phenotype. In clear cell renal cell carcinoma (ccRCC), the cancer cells utilize glutamine, leading to a local deficit in extracellular glutamine. Subsequently, cancer-infiltrating macrophages activate HIF-1α, which causes them to secrete IL-23, supporting an inflammatory phenotype [95]. Conversely, glutamine synthetase (GS) activity correlates with immunosuppressive TAM phenotypes. Metabolism via arginase-1 promotes collagen synthesis and tissue remodeling. Catabolism by indoleamine 2,3-dioxygenase (IDO) depletes local tryptophan, suppressing anti-tumor T cells [96]. In TAM-associated lipid metabolism, CD36-mediated uptake of exogenous lipids fuels OXPHOS in M2-like TAMs, supporting immunosuppression [97]. Similarly, overexpression of acetyl-CoA carboxylase (ACC) and fatty acid synthase (FASN) is linked to poor prognosis [96]. Table 1 shows metabolic plasticity across different activation states of TAMs [96,97].

Unlike conventional macrophages, TAMs exhibit context-dependent flexibility in early tumors. They show elevated glycolysis and methionine metabolism [89,97]. During progression, lactate from cancer cells reinforces their glycolytic phenotype via HIF-1α/mTOR [98]. Hypoxia drives OXPHOS/FAO dominance to sustain angiogenesis [99]. This metabolic adaptability allows TAMs to thrive in diverse TME niches, promoting immune evasion and tumor growth. Targeting pathways like LDHA (lactate dehydrogenase A) or glutaminase could disrupt their pro-tumoral functions [96].

## 4. Crosstalk Between TAMs and Tumor Cells

While macrophages within the TME accelerate tumor development by promoting proliferation, invasion, and immune suppression, these effects are orchestrated mainly through intricate crosstalk between TAMs and tumor cells. This crosstalk shapes the dynamic and pro-tumoral landscape of the TME [100].

Dendritic cells (DCs), natural killer (NK) cells, mast cells, tumor-associated neutrophils (TANs), recruited macrophages, and cancer cells are among the several cell types seen in the TME [101]. A complicated interaction between several cellular components and environmental factors causes TAM migration to the TME. Cytokines and the production of exosomes are essential components of the juxtacrine and paracrine signaling pathways that are crucial to this process. The TAM colony-stimulating factor (GM-CSF) is a chemokine that tumors secrete and essential for recruiting TAMs. Research has shown that this factor activates DCs within tumors, demonstrating anti-tumorigenic actions at low levels in the blood. On the other hand, it shifts to recruiting TAMs and promoting oncogenesis in advanced cancer stages when GM-CSF levels are high. Breast, colon, and cholangiocarcinoma cancers have all been linked to increased recruitment and infiltration of TAMs to the tumor site due to upregulation of this cytokine and CSF [18,102].

Furthermore, CSF may stimulate colon cancer cells to produce IL-8 from TAMs. Later, the colon cancer cell’s protein kinase C signaling pathway is activated by IL-8. The change causes the tumor to produce more CSF and attract more TAMs [25]. Figure 2 shows the interaction between TAMs and cancer cells. Lung cancer, osteosarcoma, and breast cancer are associated with producing IL-17, IL-34, and CSF-2, three other notable tumor-released cytokines that increase TAM recruitment [103,104]. Not only do cytokines play a crucial role in TAM recruitment, but so do specific chemokines generated by tumors. Examples include C-C motif chemokine ligand 2 (CCL2), CCL5, CCL20, CXCL4, and CXCL12. These ligands are linked to several cancers, including bladder, colon, and breast cancer and NSCLC [105,106]. In the end, reliance on these substances generated by tumors highlights how vital an inflammatory TME is for attracting TAMs, which aid in tumor development.

In addition to tumor cells, other stromal cellular components of the TME are also involved in TAM recruitment. Hepatocellular carcinoma progresses because tumor-derived CXCL5 influences TAN recruitment to the tumor site. They produce CCL2 and CCL17. These, in turn, attract TAMs, which further the cancer’s spread [18]. The production of CCL2 and other adipokines by breast cancer cells and cancer-associated adipocytes, such as lauric acid and leptin, invites TAMs to the TME [107]. Moreover, CAFs and mesenchymal stromal cells contribute to an inflammatory TME favorable to the recruitment of TAMs in ways that rely on C-C chemokine receptor type 2 (CCR2) and IL-8 [18].

A protein known as CD47 is involved in another biological pathway that regulates the interaction between tumor cells and TAMs. Tumor cells are among the many cell types that have it on their membrane surfaces. The cells produce signal regulatory protein alpha (SIRPα). It is a membrane protein that binds to CD47, and it is most prevalent in the bone marrow and TAMs (Figure 2). The usual immunoreceptor tyrosine-based inhibitory motif (ITIM) forms when these cells come together. The cytosolic tyrosine phosphatases SHP-1 or SHP-2 recruit and activate via contact between the NH2-terminal domain of the ITIM motif and the single domain of CD47. This connection regulates downstream signaling cascades and dephosphorylates many substrates, which limit TAM phagocytosis of cancer cells. The “do-not-eat-me” signal is, hence, another name for CD47 [108].

Macrophage-mediated programmed cell removal (PrCR) is essential to detect and eliminate tumors. Immune responses involving macrophage toll-like receptors (TLRs) initiate the Btk signaling pathway. In turn, it triggers phosphorylation of the endoplasmic reticulum and dissociation of the cell surface calreticulin (CRT) [18]. Macrophages express dissociated CRT, forming the CRT/CD91/C1q complexes, allowing them to phagocytose cancer cells [109]. By attaching to SIRPα on macrophages, “Do-not-eat-me” signals prevent tumor cell PrCR production, inhibiting phagocytosis. However, “do-not-eat-me” signals may be blocked by inhibiting CD47 on tumor cells. Thus, inhibiting CD47 on tumor cells and activating the TLR signaling pathway in macrophages may enhance PrCR. Research showed that tumor cells may resist macrophage phagocytosis even after CD47 inhibition. The recent literature shows a central role for major histocompatibility complex (MHC) class I in controlling the phagocytic function of macrophages and tumor defense (Figure 2). To be more precise, Weissman’s group [110] reported that tumor cells evade macrophage phagocytosis by using a different identification process between the two types of cells. To enhance phagocytosis and eliminate tumor cells in vivo, it is possible to block or downregulate the signaling protein β2-microglobulin. The protein is present on the tumor cell surface as part of MHC-I. The study showed that this therapy may significantly extend the longevity of animals for mice engrafted with MHCI^−^ tumors.

Evidence also suggests a robust relationship between TAM status and cancer stage and prognosis. In most cases, anti-tumor TAMs are often seen in early-stage tumors, while pro-tumor TAMs are more commonly found in advanced-stage tumors. Hence, the TAM polarization index, which measures the ratio of pro-tumor to anti-tumor TAMs, suggests predicting cancer outcomes and targeting potential treatments [111,112]. Upon analyzing the polarization spectra of TAMs in 931 colorectal carcinomas, Väyrynen and colleagues found that a higher M1:M2 density ratio in the tumor stroma was linked to improved cancer-specific survival probability [113]. Nevertheless, a study in glioblastoma showed that the location of these cells is just as important, if not more so, for prognosis than confirming TAMs in the TME [114].

Studying TAMs inside of the TME in a specific setting and using a particular approach might lead to substantially different percentages [115]. Several cancers, including melanoma, breast, ovarian, pancreatic, and lung cancers, are associated with a poor prognosis when these markers are present [116]. Pancreatic cancer prognosis, vascularization, and disease progression correlate with TAM levels, particularly CD163^+^ and CD204^+^ TAMs [117]. A poor prognosis and an increased risk of recurrence are linked to high levels of TAM infiltration in breast cancer [118]. Based on their research, Jeong and colleagues found that invasive breast cancer (IBC) patients had bigger tumors and a worse prognosis when CD163^+^ M2-like macrophages had infiltrated into tumor nests. This discovery also functioned as a separate indicator of prognosis for decreased disease-free survival (DFS) and OS [119]. This is mainly because, as previous research showed [120], TAMs secrete MMPs and other proteases, which aid tumor invasion and metastasis. In contrast, individuals with IBC had an improved prognosis for OS and DFS when CD11c^+^ M1-like macrophages had infiltrated the tumor stroma [85]. Mei et al. similarly discovered in their meta-analyses and systematic review that NSCLC patient’s OS improved when the tumor islet included a high density of M1-like TAMs. Still, it worsened when the tumor stroma contained a high density of M2-like TAMs [121]. The patient’s OS was unrelated to the total CD68^+^ TAMs in the tumor islet or stroma, which is intriguing [121]. These results prove that the location of M1- and M2-like TAMs, rather than just the existence of these phenotypes in the TME, is a better indicator of disease outcome.

Because TAM activities are context-dependent and highly changeable, it is essential to consider the unique TAM phenotype and its interactions with the TME when targeting TAMs for cancer treatment [122]. Cancer researchers are investigating potential ways to use TAMs as a therapeutic target. These include reprogramming TAMs to be less tumor-promoting, preventing their recruitment to tumor sites, and improving the immune response against tumors [123]. However, further research is needed to comprehend the complex interaction between macrophages and the TME fully [122,124].

## 5. Macrophages in Immunoregulation

Having discussed macrophage polarization, we now explore how these polarized TAMs influence immune regulation within the TME. Moreover, a comprehensive understanding of the crosstalk between TAMs and tumor cells highlights their role in tumor progression and underscores the broader immunoregulatory functions of macrophages within the TME. This sets the stage for exploring how these versatile immune cells orchestrate immune responses and modulate anti-tumor immunity [67,125].

In the TME, the balance between M1 pro- and M2 anti-inflammatory phenotypes in TAM determines the immune response [126]. Figure 3 shows that M2-like TAMs decrease immune surveillance and encourage tumor development, in contrast to M1-like TAMs, which enhance anti-tumor immunological responses [127]. Another method through which TAMs contribute to immunosuppression and tumor development is by releasing immunomodulatory chemicals, such as PGE2, IL-8, IL-10, and TGF-β [128]. In times of infection or inflammation, IL-10 plays an essential role in preserving tissue homeostasis by enhancing innate immunity, reducing excessive inflammatory responses, and facilitating processes for tissue repair [129]. When they are present with cancer, they suppress the functions of other immune cells, such as NK and T cells. This helps tumor cells survive by avoiding immune surveillance [130]. Recent research has shown that TAMs influence NK cell cytolytic activity through two mechanisms. Through a contact-dependent mechanism mainly mediated by TGF-β, TAMs reduce the cytolytic activity of NK cells. The restoration of NK cell cytotoxicity suppresses TGF-β. Conversely, TAMs encourage the CD27^low^CD11^high^ NK cell phenotype, characterized by reduced activation and tumor-killing capabilities. This adds to NK cell exhaustion [131]. By activating Foxp3, TAMs convert Th cells into Tregs in response to the release of TGF-β [132,133]. TAMs activate Tregs via the release of TGF-γ and IL-10, and they hinder the activity of CD8^+^ effector T cells. Lastly, TAMs secrete chemokines, such as CCL5, CCL17, CCL20, and CCL22, which attract CCR4^+^ Tregs and help them enter the TME [102,134]. The immune suppression of CD8^+^ cytotoxic T cells is intensified as a result.

In addition, the beneficial interaction between TAMs and Tregs has been extensively studied. By aiding in the formation of an immunosuppressive TME, Tregs contribute to immune evasion in cancer. Research has shown that TAMs may activate Tregs, encouraging monocytes to differentiate into M2 phenotype cells [135]. Clinical trial results show that M2-like immunosuppressive TAM infiltration at metastases suppresses clinically significant immune responses in the metastatic TME (M-TME). Patients without neoadjuvant treatment for high-grade serous carcinomas (HGSCs) had their transcriptome profile examined across 24 matched original and metastatic tumor tissues. The study found that M2-like TAMs may cause T cell exhaustion in the M-TME because they regulate the expression of cytokine/chemokine signals, such as IL-10 and CCL22. Metastatic site inhibition of immune responses was associated with poor prognoses in HGSC patients with robust M-TME infiltration by M2-like TAMs. Immune effector cell infiltration did not significantly affect patient survival, and 1468 genes showed differential expression between the primary-TME and M-TME of HGSCs [136].

In addition, TGF-β impacts DCs, and TAMs produce IL-10. Evidence shows that these anti-inflammatory cytokines cause antigen-presenting cells to die off. This reduces the number of DCs that can infiltrate tumor metastasized sites and migrate to lymph nodes, weakening adaptive immune responses mediated by T cells [137,138].

Another way that TAMs might limit the immune response is through their interactions with other immune cells. A critical component of immunosuppression is programmed PD-1, which belongs to the CD28 superfamily. When regulating the immune system for objectives including cancer prevention, infection control, autoimmune disease treatment, and organ transplant recipient survival, PD-1 consideration is essential [139,140]. PD-L1 is a PD-1 ligand produced by antigen-presenting cells. Collaborating with PD-1 on T cells stops T cells from targeting antigen-presenting cells [139,140]. The PD-1/L1 signaling pathway can restrict the actions of T effector cells, DCs, and NK cells. As shown by the reduction of phagocytosis in TAMs and the suppression of effects on T cell activation, proliferation, and cytokine production, this limitation increases the possibility of tumor immune escape (Figure 3) [141,142].

To specifically decrease the immune response, TAMs may increase the expression of additional surface proteins, including CD80/CD86 or death receptor ligands like Fas-L or TRAIL [143]. These ligands bind to and activate inhibitory receptors on immune effector cells, namely CTLA-4, FAS, and TRAIL-RI/-RII. Stimulating the PD-1 and CTLA-4 receptors reduces cytokine and protein production, which aids cell survival by inhibiting the T cell receptor (TCR) signaling pathway. On top of that, TAMs produce the enzyme Arg-1, which is responsible for L-arginine degradation. Several processes rely on L-arginine, including developing immunological memory, lymphocyte proliferation, TCR complex expression, and the anti-tumor response mediated by T cells [18,144,145]. Therefore, TAMs reduce adaptive immunity against tumors by reducing immune responses pleiotropically.

Anergic and unresponsive T cells, as opposed to activated ones, might result from defective immunological synapses during the presentation of antigen fragments by macrophages to T cells [146]. The M2-induced Treg population maintains this T cell anergy condition, most noticeably in the lymph nodes that drain tumors [147]. Moreover, Kersten and colleagues show that CD8^+^ T cells are prepared to saturate following extended, antigen-specific engagement with TAMs, and this saturation amplifies hypoxic settings, such as the TME [148]. This immune-excluded TME pattern further enhances persistent contact with TAMs, inhibiting CD8^+^ T cell tumor invasion. By combining CSF-1R blockage with anti-PD-1 treatment, Peranzoni and colleagues demonstrated that CD8^+^ T cell tumor infiltration improved, and tumor growth was delayed [149].

**Figure 3 cells-14-00741-f003:**
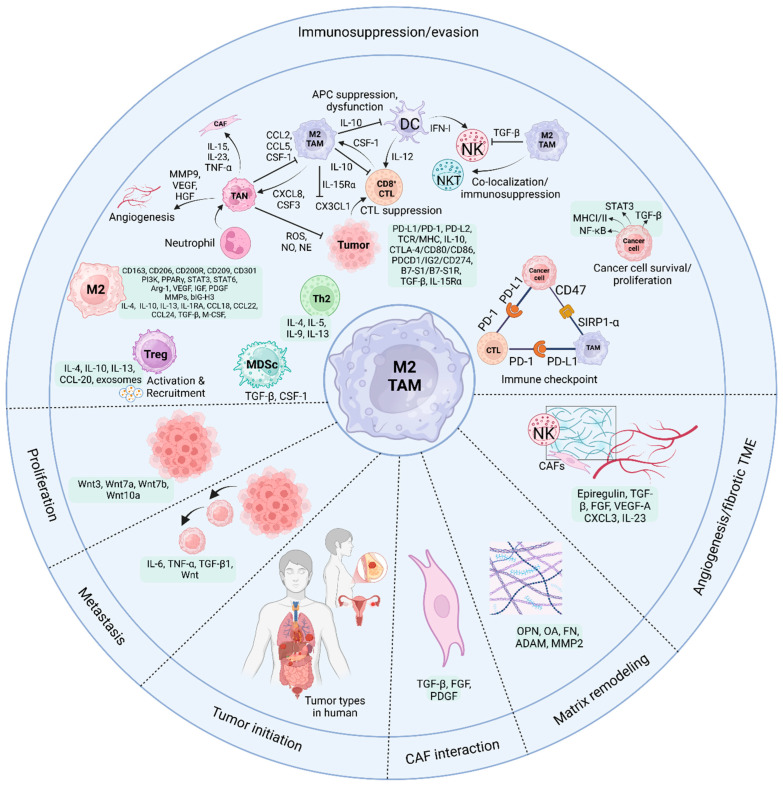
The TAMs regulate tumors and polarize M1/M2 cells. The tumor stage, tissue, microenvironment, and paracrine signals affect them. Macrophages recruit the support of T and NK cells to fight against cancers. Cancer and tissue repair are two macrophage populations that produce antigen-presenting cells (APCs) like M1. Malignant M2 macrophages cause tumors to develop. Macrophages in the hypoxic TME use immunosuppressive mediators to fuel tumor development, angiogenesis, invasion, and metastasis. The same holds for inflammation, cancer stem cell (CTC) self-renewal, tissue remodeling, and EMT. Moreover, to coordinate CD8^+^ T cell responses, TAMs activate immunological checkpoints, downregulate antigen presentation, and release regulatory factors. Additionally, TAMs inhibit dendritic cell antigen presentation and invasion. TAMs release TGF-β and CSF-1, which boost myeloid-derived suppressor cell (MDSC) proliferation. TAMs attract immune-suppressive Treg cells and impair NKT cell activity, suppressing it. Furthermore, PD-L1 on TAMs and tumor cells interacts with PD-1 on cytotoxic T lymphocytes (CTLs) to reduce anti-tumor immunity. CD47-SIRPa interaction between TAMs and cancer cells helps tumors avoid phagocytosis. Moreover, chemokines, such as CCL1, CCL17, CCL18, CCL22, and CCL24, are abundantly expressed by M2-like macrophages, along with CD163, CD206, CD200R, CD209, and CD301. Various anti-inflammatory factors, such as TGF-β, IL-4, IL-13, IL-10, and IL-1RA, are released through macrophages. Furthermore, M2-like TAMs have reduced the amounts of the inflammatory cytokines IL-6, IL-12, IL-23, and TNF-α. In addition to MMPs, M2-like macrophages secrete autocrine extracellular matrix components, including fibronectin, beating-h3 (BIG-H3), enzymes that cross-link ECM, transglutaminase, and proteins that bridge gaps in bone [150]. M2-like macrophages increase Arg-1 and VEGF expression, aiding proline and polyamine production. Proline promotes ECM formation, and polyamines enhance cell growth. PDGF and IGF, which enhance cell proliferation, are also released by M2-like macrophages and implicated in angiogenesis [151,152]. Created in BioRender. Saeed, A. (2025) https://BioRender.com/kim50ti.

### 5.1. TAMs and CD8^+^ CTLs

CD8 T cells, also known as CTLs, are a specialized subset of T lymphocytes (white blood cells) that play a central role in the immune system’s defense against intracellular pathogens, such as viruses and some bacteria, as well as cancer cells [153]. Among the many immunosuppressive components of the TME, TAMs have a role in tumor initiation, progression, and immunotherapy resistance. Cancer development is accompanied by fibrosis, which reduces anti-tumor immune infiltration. Several factors reduce the anti-tumor responses of CD8^+^ T lymphocytes by lowering their effector activity [154].

A recent study reported that in response to the solid fibrotic TME, TAMs initiate collagen production via TGF-β. Collagen-synthesizing macrophages absorb ambient arginine, manufacture proline, and release ornithine, which impairs CD8^+^ T cell activity in female breast cancer. Thus, a solid and fibrotic TME may inhibit anti-tumor immunity by physically excluding CD8^+^ T cells and mechano-metabolic programming TAMs, generating an unfavorable metabolic milieu for anti-cancer immunotherapies [155].

Few TAM-targeting drugs work preclinically or clinically. To maintain CSF-1 receptor inhibition and high anti-cancer immunity, the novel reagent FF-10101 formed a covalent link and decreased immunosuppressive TAMs in the TME. Preclinical animal studies showed that FF-10101 decreased immunosuppressive TAMs and boosted TME anti-cancer TAMs. CD8^+^ T lymphocytes that target tumor antigens also increased, inhibiting tumor development. FF-10101 with an anti-PD-1 antibody destroyed tumors better than either alone. As in animal models, FF-10101 reduced TAM PD-L1 expression in human cancer tissues. By reducing immunosuppressive TAMs and enhancing tumor antigen-specific T cell responses, FF-10101 enhances TME. FF-10101 may work with PD-1/PD-L1 immune checkpoint inhibitors in cancer immunotherapy [156].

Xie et al. [157] found E-twenty-six variant transcription factor 4 (ETV4), which elevated PD-L1 and chemokine CCL2 production in HCC cells. This, in turn, led to the accumulation of TAM and MDSC and the suppression of CD8^+^ T cells, which facilitated the spread of HCC [158]. In another study, Luan et al. [157] found aberrant complement 5a receptor (C5aR) expression in human ovarian cancer (OC) and elevated C5aR expression on TAMs, which activated TAMs to become immunosuppressive. C5aR deletion or inhibitor therapy mitigated tumor development and restored the TAM anti-tumor response. Mechanistically, C5aR deficiency reprogrammed macrophages from pro-tumor to anti-tumor, upregulating immune response and stimulation pathways. Consequently, it enhanced CTL’s anti-tumor response in a manner dependent on CXCL9. Drugs that inhibit C5aR also enhance immune checkpoint blockage. C5aR expression is linked to CXCL9 production and CD8^+^ T cell infiltration, and a high C5aR level indicates worse clinical outcomes and lower anti-PD-1 treatment benefits. The findings illuminate how the C5a–C5aR axis modulates TAM anti-tumor immune response and suggest therapeutic uses for targeting C5aR.

Bader et al. [159] reported that obesity increased TAM PD-1. When exposed to type I inflammatory cytokines and obesity-related substances, such as IFN-γ, TNF, leptin, insulin, and palmitate, macrophages express PD-1 via mTORC1 and a glycolysis-dependent approach. PD-1 negative feedback reduced TAM glycolysis, phagocytosis, and T cell activation. PD-1 inhibition increased macrophage glycolysis necessary for TAM CD86, MHC I and II, and T cell activation. Myeloid cell PD-1 deficiency reduced tumor growth, increased TAM glycolysis and antigen presentation, and exhausted CD8^+^ T cells. Inflammatory stresses and an obesity-associated rise in metabolic signaling lead to TAM PD-1 expression, creating a TAM-specific feedback loop that suppresses tumor immune surveillance. It may boost PD-1 therapy but increase obesity-related cancer risk.

In another study, lentiviral vectors (LVs) generated liver macrophages, including Kupffer cells and TAMs, that transport IFNα to liver metastases. IFNα gene therapy reduces liver metastases in colorectal and pancreatic ductal adenocarcinoma mice. TAM immunological activation, MHC-II-restricted antigen presentation, and reduced CD8^+^ T cell exhaustion result from IFNα response. IL-10 signaling, Eomes CD4^+^ T cell proliferation, Tr1 cell traits, and CTLA-4 expression increase with treatment resistance. CTLA-4 immune checkpoint inhibition and IFNα LV infusion boost tumor-reactive T cells, leading to a complete response in most animals. These data suggest an unmet medical requirement for therapy [160].

### 5.2. TAMs and NK Cells

NK cells are a specialized type of white blood cell and a key component of the innate immune system. As a subset of lymphocytes, alongside B and T cells, NK cells play a crucial role in the body’s first line of defense against infections and cancer [161]. The innate and adaptive immune systems interact via macrophages and phagocytic and antigen-presenting cells. There are two kinds of TAM: TAM1 and TAM2. TAM1 has anti-tumor action, whereas TAM2 promotes tumor growth. IFN-γ and LPS trigger TAM1 induction, producing cytokines including TNF-α, IL-1β, IL-6, IL-12, nitric oxide synthase (iNOS), and reactive oxygen species (ROS). In response to cytokines like IL-4, IL-10, and IL-13, TAM2 is activated, generating cytokines like IL-10, TGF-β, and VEFG [161,162].

Initially, TAM1 enhances the expression of CD69, an activation marker, degranulation, and IFN-γ secretion of resting autologous NK cells through both soluble factor secretion and direct interaction [161,163]. When TAM produces IFN-γ, it triggers the expression of IL-15Rα on NK cells and the subsequent production of IL-15 [164,165]. NK cells produce more IFN-γ due to this cis-presentation. The release of IL-1β, IL-23, and IFN-β, as well as the upregulation of *NKG2D* and *NKp44* expression, are mechanisms via which TAM1 enhances the cytolytic efficacy of NK cells [161]. Furthermore, the cytotoxicity of murine splenic CD49b^+^ NK cells and their increased *NKG2D* expression against several tumor cell lines are both enhanced by macrophages treated with polyI:C. The inhibitory receptor *NKG2A* identifies Qa-1, which these murine macrophages express, allowing them to evade NK cell death [163,166]. IL-12 family cytokines are produced by monocyte-derived macrophages cultured with IFN-λ/LPS, which stimulates the production of IFN-γ by human NK cells [167].

In one investigation, researchers examined how DCs and NK cells work together to eliminate tumors. One receptor that macrophages and DCs produce, Dectin-1, can recognize N-glycan complexes in tumor cells. This triggers the activation of IRF5, leading to gene transcription downstream. Due to these processes, macrophages and DCs can trigger NK cell anti-tumor activity against tumor cells that produce N-glycan [168].

On the other hand, TGF-β produced from TAM hinders NK cell’s abilities by reducing cytokine production and degranulation in humans [169] and cytotoxicity in mice [170,171]. TAM also promotes the exhausted phenotype CD27^low^ CD11b^high^ in mice [171]. Targeting VEFG-A in myeloid cells enhances chemotherapy transport to the tumor nest and subsequent NK cell recruitment, as shown in a mouse model by Klose et al. [172].

NK cells can impair macrophages’ ability to fight tumors. An exhausted phenotype, defined by a rise in PD-1 and TIM-3 expression and a reduction in *NKG2D*, is seen, for instance, in prostate cancer patients’ peripheral blood NK cells [173]. Two inhibitory surface molecules worth mentioning are PD-1 and TIM-3. When PD-1 and its ligands, PD-L1 and PD-L2, are overexpressed in some malignancies, they decrease the anti-tumor activity of NK [173,174]. Patients with prostate cancer consistently have NK cells with a significantly reduced degranulation capability [131]. Figure 3 shows that these NK cells release soluble mediators, such as CCL2 and IL-10, which play a role in monocyte recruitment and M2 polarization [175].

### 5.3. TAMs and NKT Cells

Natural killer T (NKT) cells are a unique subset of immune cells that bridge the innate and adaptive arms of the immune system. Features of both conventional T lymphocytes and NK cells characterize them [163,176]. TME NKT cell investigations have increased in recent years. NKTs with decreased CD1d expression may be classified into many types, including Th1-like, Th2-like, Th17-like, Treg-like, and T-follicle-assisted (TFH)-like [177]. NKTs attack tumor cells like NK cells but flip between inflammatory and immunosuppressive characteristics. The central subgroup explored is invariant natural killer T (iNKT). iNKT cells possess inherent characteristics and secrete several cytokines from the Th1 and Th2 groups, such as IFN-γ, IL-4, and GM-CSF, which may control activated APCs [178]. High cytolytic activity, the release of cytotoxic particles, including perforin and granzymes, and the activation of death receptor pathways involved in the Fas-FasL-TRAIL-DR5 connection were additional features of iNKT cells, which were unique from other cell types [179].

Recent studies show that crosstalk between iNKTs and macrophages helps better understand immune cell interaction in the TME. While CD68^+^ macrophages enhanced periampullary malignant adenoma survival, co-localization of NKp46^+^ NKT cells with CD163^+^ macrophages reduced it [180,181]. Although NKp46^+^ NKT cells are abundant in the tumor region close to tumor cells, CD68^+^ macrophages are primarily found in the stroma. They are located far from tumor cells in malignant tumor tissues. This limits their capacity to interact [181]. Immunosuppressive macrophage co-localization reduces NKT function (Figure 3). A current study shows that iNKT cells control TAMs. Using CD1d and Fas-FasL, iNKT cells kill M2 TAMs, whereas M1 TAMs with CD40 are protected [182]. A mechanism involving microsomal prostaglandin E synthase-1 (mPGES-1) and 5-lipoxygenase (5-LOX) is proposed by iNKT cells to inhibit M2 TAMs in the pancreatic cancer model [183]. However, iNKT cells improved TAMS M2 polarization, FoxP3 protein expression, Treg frequency, tumor growth, and intestinal adenomatous polyps in colon cancer transgenic mice [177,184]. Few studies have examined the interaction between NKT cells and TAMs, which may vary significantly amongst tumors. More research on the process is needed.

A potential cell platform for CAR treatment in solid malignancies is human NKT cells. Li et al. [185] recently generated universal CAR-engineered NKT (^U^CAR-NKT) cells that selectively deplete immunosuppressive TAMs to change the TME. ^U^CAR-NKT cells also have fewer challenges with cytokine release syndrome and graft-versus-host disease. To pave the way for the translational and clinical development of ^U^CAR-NKT cell products, preclinical investigations prove their viability and therapeutic potential.

In another study, researchers Zhou et al. [186] discovered that CAR-NKT cells can eliminate M2-like macrophages that express CD1d. Additionally, CAR-NKT cells stimulate endogenous T cells to combat neoantigens linked with tumors and spread epitopes. The findings assist the clinical development of CAR-NKT treatments by showing their multimodal action in solid tumors.

### 5.4. TAMs and CD4 T Cells

CD4 T cells, also known as CD4^+^ T cells or T helper cells, are a specialized subset of white blood cells that play a central role in coordinating the body’s immune response against infections and diseases. They are called “CD4” because they express the CD4 glycoprotein on their surface. It is a co-receptor for the T cell receptor (TCR) and crucial for recognizing antigens and other immune cells [187]. Current research is focused on CD4^+^ T cell function in tumor immune evasion and anti-tumor immunity. CD4^+^ T cells vary like macrophages. Effective CD4^+^ T cells are regulatory (iTreg and nTreg) or helper (Th1, Th2, Th17, Th9, and Th22) in the TME [188,189]. Heterogeneity casts doubt on cancer immunity. CD4^+^ T cells cannot recognize most cancer cells because they lack MHC class-II molecules (MHC-II) or HLA-DR. Necrotic tumor cells or cancer cell vesicles eaten by tumor stromal cells usually cross-present tumor antigens using the classical MHC-II processing pathway [190,191]. MHC-II-positive monocytes/macrophages are found in most solid tumors. Therefore, CD4^+^ T cell immunological recognition relies on them. TAMs increase MHC-II genes and transmit antigens to CD4^+^ T cells in early NSCLC lesions [192]. Curiously, one study discovered that CXCL13^+^ CD4^+^ T cells are encouraged to invade by activated macrophages via IL15, CXCL9, CXCL10, and CXCL11. Only in the microenvironment does this percentage of T cells boost melanoma patient survival [133]. The elimination of MHC deletion and non-responsive cancer cells is triggered by inflammatory cell death. This is initiated by CD4^+^ effector T cells and activated iNOS-expressing tumor-killer monocytes and macrophages [193]. There is a distinct opportunity for targeted treatment targeting this small subset of CD4^+^ effector T cells.

Macrophage–CD4^+^ T cell interactions may not always kill tumor cells. In cirrhosis, macrophage-induced CD4^+^ T cells may become TREGs, leading to immunosuppressive TME and HCC [194]. Additionally, TAMs increased Treg cell proliferation and the Treg/CD8^+^ T cell ratio [111]. In addition to expressing PD-L1 on the cell membrane, TAM in the tumor’s core may recruit invasive Tregs from the tumor’s periphery into the TME. Therefore, it suppresses the immune response (Figure 3) [195,196]. Around tumor cells in stomach cancer models, you can find CD4^+^FoxP3^−^ T cells, CTLA-4 T cells, and PD-L1 T cells. However, to successfully transfer antigens, CD68^+^CD163^−^HLA-DR^+^ (M1) macrophages are too far away [197]. Patients with head and neck squamous cell carcinoma with PD-1^+^ Th cells and CD163^+^ TAMs outlived those in other subpopulations [198].

Regarding tumor development, immunological homeostasis, and regulating other immune cell functions, macrophages and Tregs play a pivotal role. New research shows that metabolic alterations in Tregs and macrophages regulate signaling cascades and epigenetic reprogramming, affecting their pro- and anti-cancer activities. So, they are acknowledged as potential targets for cancer immunotherapy more and more. Some metabolites in the TME may influence their pro- or anti-tumor activities via metabolic machinery interference [199]. Zhang et al. found that colorectal cancer included two main TAM subgroups: C1QC^+^ and SPP1^+^. The first subtype is associated with tumor angiogenesis and vascular markers, whereas the second subtype upregulates genes involved in complement activation and antigen presentation [200]. Another study found that GITR^+^CD25^+^ effector Tregs, which express the scavenger receptor CD36, are the most suppressive subgroup of Tregs in human melanoma tumors [201].

### 5.5. TAMs and DCs

Dendritic cells (DCs) are specialized immune cells called APCs. Their primary function is to process antigens, molecules from pathogens, or damaged cells and present them on their surface to T cells, thereby initiating and shaping the adaptive immune response. Standard dendritic cells are cDC and pDC [202]. Due to their antigen-capturing capabilities, the cDC1s favor MHC-I cross-presentation on CD8+ T cells, whereas the cDC2s opt for MHC-II on CD4^+^ T cells [203]. In addition, TLRs activate pDC, a DC subgroup, to release vast amounts of type I IFNs (IFNs) to fight cancer. Precisely activated pDC stimulates T cells in vitro [204]. In vivo studies demonstrate that pDC kills cancer [205]. According to a study, pDC is associated with poor clinical outcomes caused by tumor suppression resistance and has unfavorable immunomodulatory effects on the TME [106]. Regulatory pDCs lack IFNs and downregulate costimulatory molecules, while IDO and PD-L1 upregulate [206].

The extra-tumor stroma is where PD-L1^+^ DC and TAMs are most often seen in esophageal cancer patients. Their prognosis is dismal, according to recent studies [207,208]. Macrophage–DC co-localization may inhibit DC invasion or antigen presentation. In the B78ChOVA melanoma model, TAMs were more invasive, and CD103^+^ cDC1 and CD11b^+^ cDC2 decreased [177]. Preventing antigen presentation may inhibit CD4^+^ and CD8^+^ T cell activation. TAM suppresses CD103^+^ DC IL-12 production and T cell activation in breast cancer by secreting IL-10 (Figure 3) [209]. Unfortunately, previous studies have revealed that the TAM–DC connection does not improve tumor prognosis. Little is known about this interaction, so the mechanism is uncertain and needs further attention.

### 5.6. TAMs and Neutrophils

Neutrophils are the immune cells that first combat infection and inflammation. Vascular chemotaxis transports them into tissues to fight infections. TME neutrophil activation and function vary based on numerous parameters. These neutrophils become tumor-linked. Like TAMs, tumor-associated neutrophils (TANs) have two main polarizing types: N1TANs, which fight cancer, and N2TANs, which promote it. Many studies have studied TAN’s anti-cancer effects. TANs penetrate cancer cells, expressing co-stimulatory receptors, such as 4-1BBL, OX40L, and CD86. They activate T cells and secrete IFN-γ for anti-tumor activity [121]. Interestingly, IFN-γ activates NK cells by causing TANs to generate IL-18 [210,211]. Furthermore, DC and CD8^+^ T lymphocytes are stimulated by TAN’s production of TNF-α [212]. TANs may kill cancer cells by secreting ROS, NO, and NE, reducing tumor growth and metastasis [213]. TANs can potentially enhance tumor development by increasing cancer cell proliferation, invasion, angiogenesis, and immunosuppression. The Akt/p38 pathway is activated by TANs, which turn MSCs into TAFs and promote tumor cell proliferation and metastasis via the production of cytokines, such as IL-17, IL-23, and TNF-α [214,215]. Like macrophages, TANs increase angiogenesis and cancer cell aggressiveness by producing VEGF, HGF, and MMP9 [216,217]. After G-CSF and TGF-β stimulation, TANs release arginine-1, reactive oxygen species, and nitric oxide, which limit T cell activation [218,219].

In a study of intrahepatic cholangiocarcinoma, most CD66b^+^ TANs were near CD68^+^ TAMs and formed small cell clusters in two-thirds of the samples [220]. TANs-TAMs clusters boosted HuCCT1, RBE, and SG231 cell proliferation, invasion, colony formation, and lung metastatic tumors compared to TANs or TAMs alone [78,221]. TANs preferentially create OSM (oncostatin M), and TAMs preferentially express IL-11, which promotes STAT3 signaling in ICC cells and tumor development [177,220]. TANs release CCL2, CCL5, and CSF1, which may recruit macrophages to TAMs [150]. TAMs release CXCL8 and CSF3, TAN-related chemokines [150]. A positive feedback loop between TANs and TAMs boosts CSF1 and CXCL8 secretion after co-culture, substantially overlapping geographically and synergistically (Figure 3).

Recently, it was found that CD89^hi^CD32^lo^CD64^lo^ peripheral blood neutrophils (PBNs) and TAN supported tumor cell development in the presence of cetuximab. In contrast, IgA anti-EGFR Abs produced PBN tumoricidal and removed TAN’s stimulatory action. This work illuminates how myeloid effectors kill or resist tumor cells during tAb treatment [222]. Lei et al. [223] showed CXCL5-induced neutrophil NETosis. CD8^+^ T cells treated with neutrophil extracellular traps (NETs) upregulated exhaustion-related and cytosolic DNA sensing pathways. Likewise, they downregulated effector genes. However, A2AR inhibition dramatically decreased CXCL5 expression and neutrophil infiltration, improving CD8^+^ T cell dysfunction. According to the results, the complicated connection between tumor and immune cells may be a therapeutic target. Schmidt et al. [224] identified poor prognoses in central tumor samples with high stromal and intraepithelial CD66b^+^ TAN density. Higher neutrophil density in lymph nodes and adjacent normal breast tissue decreased disease-free survival. They previously linked TAN density to CD163^+^ M2-like TAM density. Low M1/M2 TAM ratios did not negatively prognosticate TANs, whereas high ratios did. The TAM polarization state was the only independent predictor in a multivariate TAM and TAN density study. In single-marker analysis of early-stage luminal breast cancer, CD66b^+^ neutrophils were adverse. Future studies may require TAM analysis to estimate their predictive impact correctly. Malignancies that cannot attract and polarize TAMs may compensate for immunoevasion and disease progression by recruiting TANs. Pan et al. [225] showed that TAMs lacking VSIG4 produced less lactate and histone H3 lysine 18 lactylation. Furthermore, it lowered *STAT3*-mediated transcription of SPP1, disrupting TAM–neutrophil cell–cell connections. VSIG4 and SPP1 inhibition synergistically increased anti-tumor action. According to the study, *VSIG4’s* epigenetic regulatory function allows TAMs to construct the immunosuppressive TME. Likewise, it hinders antigen-specific immunity against aggressive tumors and their checkpoint role.

### 5.7. TAMs and MDSCs

Myeloid-derived suppressor cells (MDSCs) are a heterogeneous group of immune cells originating from the myeloid lineage, a family of cells that develop from bone marrow stem cells. Bone marrow MDSCs produce DC, macrophages, and granulocytes. To create the immunosuppressive tumor myeloid microenvironment, CCL2 and CCL5 draw it to the tumor center [196]. The positive feedback impact of TAM secreting TGF-β has been shown in recent research. The amplification of MDSCs might be enhanced by continuous exposure to TGF-β and CSF-1 [226]. TAMs and MDSCs co-localize at the aggressive border [227]. HCC also demonstrated TAM and MDSC-induced CD8^+^ T cell suppression along the tumor border [228,229].

Furthermore, the TRAMP/MICB spontaneous prostate tumor model’s flow cytometry revealed a high association between MDSCs in the tumor-infiltrating region and serum soluble MHCI chain-related molecules (SMIC). It is an NKG2D ligand that activates *STAT3*, increases MDSCs, and polarizes TAMs M2 [230]. Hypoxia improved sialic acid transport and CD45 binding when recruited MDSCs moved to tumors, activated CD45 protein tyrosine phosphorylase, dephosphorylated, and downregulated *STAT3*. It accelerated TAM differentiation without HIF-1 [231,232]. Though paradoxical, tumor tissues may up- and downregulate myeloid cell *STAT3* activity due to time and space. The above method may upregulate *STAT3* to enhance MDSCs from blood vessels entering the tumor tissue. When recruited and amplified, MDSCs penetrated tumor tissue in the deep vascular deficit area and hypoxia downregulated STAT3 and expedited MDSC differentiation into TAMs. This positive space–time feedback may assist macrophages and MDSC in growing tumors.

Xie et al. [158] showed in HCC cells that overexpression of ETV4 stimulated PD-L1 and CCL2 expression, leading to an upregulation of TAM and MDSC infiltration and a downregulation of CD8^+^ T cell accumulation. Lentivirus- or CCR2-inhibitor CCX872-mediated CCL2 knockdown reduced ETV4-induced TAM, MDSC infiltration, and HCC metastasis. In addition, the ERK1/2 pathway is used by both FGF19/FGFR4 and HGF/c-MET to enhance ETV4 expression. Moreover, a positive feedback loop was established between FGF19, ETV4, and FGFR4 when ETV4 increased FGFR4 expression, and FGFR4 downregulation reduced ETV4-enhanced HCC metastasis. Finally, anti-PD-L1, in conjunction with either the FGFR4 inhibitor BLU-554 or the MAPK inhibitor trametinib, significantly reduced the spread of HCC produced by the FGF19-ETV4 signaling pathway. The authors conclude that an effective strategy to suppress HCC metastasis might be using anti-PD-L1 paired with the FGFR4 inhibitor BLU-554 or the MAPK inhibitor trametinib, as ETV4 is a predictive biomarker.

Using scRNA-seq analysis, Tang et al. [233] identified two states of macrophage cells, MDSC-like (expressing CD300E, VCAN, EGFR, FCN1, and CCL20) and TAM-like (expressing C1QA, APOE, CD163, MRC1, and FOLR2). Additionally, they discovered that macrophages resembling MDSCs and TAMs increased levels of proteins that inhibit the immune system and promote cancer development, including SPP1, CCL20, and TIMP1. Additionally, increased MDSC- and TAM-like macrophages indicate an immunosuppressive milieu that promotes tumor development and immune evasion. Immunosuppressive myeloid cells, including TAMs and MDSCs, and immune-inhibitory factors may reduce the effectiveness of ICB in gastric cancer liver metastasis. Findings from the research highlight the need to investigate the liver microenvironment for tumor cell and immune system interactions to improve liver metastasis therapies, circumvent immune resistance, and improve patient outcomes. Tabachnick-Cherny et al. [234] used a single-cell technique to investigate MCC myeloid signatures. The authors then confirmed markers in tumor samples before PD-L1 blockade using myeloid spatial biology. Researchers identified and characterized MCC’s dominant myeloid cells, TAMs, which show M-MDSC traits, such as increased S100A8 and S100A9 genes and immunosuppressive cytokines like IL-10 and VEGF.

Zhang et al. [235] showed that C3 of RCC cell-derived EVs aids metastasis by attracting polymorphonuclear (PMN)-MDSCs and polarizing TAMs into an immunosuppressive phenotype. From a mechanistic standpoint, EV C3 increased TAM polarization and PMN-MDSC recruitment by inducing lung macrophages to secrete CCL2 and CXCL1. Significantly, in a mouse model of RCC-induced C3-induced lung metastasis, the inhibitors RS504393 and Navarixin, which target the CCL2/CCR2 or CXCL1/CXCR2 axis, respectively, successfully decreased metastasis. From a clinical standpoint, the prognosis is not suitable for RCC patients who exhibit a high level of C3. They found that RCC metastasis is facilitated by tumor-derived EV C3, which produces an immunosuppressive TME via TAMs.

### 5.8. TAMs and γδ T Cells

The subgroup of T lymphocytes known as γδ T cells, which contribute to the immune cells that infiltrate tumors, constitutes 1% to 10% of the T cells in the peripheral blood of healthy individuals [236]. These T cells express T cell receptors (TCRs) with γ and δ chains, distinguishing them from those with α and β chains [236]. While most T cells in the body express αβ TCRs, γδ T cells have recently come to the forefront due to their unique characteristics, which make them promising targets in cancer immunotherapy [237]. γδ T cells deliver antigens without the aid of MHC, making them unique. This is beneficial for immunotherapy, as standard T-cell-based treatments are hampered by cancer cells’ ability to downregulate MHC-I expression to escape detection [238].

The results demonstrated that lipid formulations increased circulating γδ T cells via MPS cells and suppressed TAMs in a breast cancer mouse model [239]. TAMs may be responsible for the reported anti-tumor activity in mice models lacking γδ T cells by endocytosing n-BPs, which have been proven to deplete these cells [239]. The M1: M2 TAMs ratio in tumor models suggests that liposomal ZOL may preferentially remove M2 macrophages. However, it may also remove M1 anti-tumoral and M2 pro-tumoral TAMs in hepatocellular carcinoma and triple-negative breast cancer xenograft mouse models [240,241,242]. Adoptive immunotherapy using liposomal ALD-activated Vγ9Vδ2 T cells effectively treats mouse epithelial OC [243,244]. In laboratory and animal studies, nanocarriers of lipids have shown excellent efficacy in stimulating γδ T-cell-mediated lysis of cancer cells. Still, toxicity and bioavailability difficulties remain in attaining therapeutic usefulness. The study by Gao et al. [245] demonstrated that BTNL2 stimulates γδT17 cells to release IL-17A and bring in myeloid suppressor cells, leading to a rise in the quantity of TME mesenchymal stem cells, M2 macrophages, and TAMs.

## 6. Immunotherapy Employing Macrophages and Anti-PD-1/PD-L1

Establishing macrophages’ pivotal roles in immunoregulation, recent advances have focused on harnessing these cells in immunotherapeutic strategies, particularly through interventions targeting the PD-1/PD-L1 axis, which modulates macrophage function and enhances anti-tumor immune responses [67,134].

Several immune checkpoint blockade therapies are available, but the two most common are anti-PD-1 and anti-PD-L1 treatments. Cancer immunotherapy that inhibits PD-1 aims to counteract immune suppression rather than boost immunity. Inhibitors of the PD-1/PD-L1 pathway have enhanced the cytotoxicity of T cells and the treatment of cancer [246]. As a means of immune system evasion and inhibition, TAMs express PD-L1 and PD-1. Inhibiting PD-1/PD-L1 may restore PD-1^+^ TAM phagocytosis, which in turn reduces tumor burden [247].

### 6.1. Effects of TAMs on PD-1/PD-L1 Expression

Immunotherapy targeting PD-1/PD-L1 has been used or attempted in the treatment of many solid tumors, such as lung cancer, advanced metastatic melanoma, esophageal cancer, and colorectal cancer [248,249]. The PD-1/PD-L1 pathway was aberrantly activated in several cancers [13,250]. Nevertheless, PD-1 inhibitors failed to improve outcomes for many patients despite strong PD-L1 expression, and the exact reasons behind this are still poorly understood.

As previously shown in studies, TAMs activate many signaling pathways that influence PD-1/PD-L1 expression, which impacts the effectiveness of PD-1/PD-L1 inhibitors. There is a positive connection between CD163^+^ TAMs in the TME and PD-L1 expression in many tumor types, including pancreatic and liver tumors. Multiple signaling pathways being activated, such as phosphoinositide 3-kinase (PI3K)/AKT, NF-κB, Janus kinase (JAK)/*STAT3*, or Extracellular signal-regulated kinase (ERK) 1 and 2, may lead to upregulation of PD-L1 expression by various cytokines generated by TAM, such as IL-6 and TNF-α [251,252]. Moreover, TNF-α may potentially upregulate PD-L1 expression of proteins through post-translational regulation [251].

### 6.2. TAMs and Resistance to Anti-PD-1

Several factors have been associated with anti-PD-1 resistance, including TME and PD-L1 expression on tumor cells. As mentioned earlier, regulating PD-L1 protein expression through TAMs’ cytokine is a notable indicator for anti-PD-1/PD-L1 treatment. The TME now contains many immune cells, and cancer ecology has changed over the years, intricately impacting cancer growth [200,253]. The response to immunotherapy was associated with the interaction between macrophages and other immune cells [253]. Researchers found that immunotherapy is less effective when FAP^+^ fibroblasts and SPP1^+^ macrophages interact, leading to the development of immune-excluded desmoplastic structures and a restriction of T cells [253]. The therapeutic benefit of checkpoint inhibitors may be predicted in triple-negative breast cancer patients with high numbers of CXCL13^+^ T cells. These cells are linked to macrophages’ pro-inflammatory characteristics [200].

A few biological processes in cancer rely on exosomes, which are tiny extracellular vesicles. According to recent research, metastasis and tumor development are supported by pre-metastatic niches, which may enhance exosomes produced by macrophages. Depending on their source, EVs produced by M2 macrophages may influence the immune cell spectrum in the TME. They induce resistance to anti-PD-1/PD-L1 treatment or increase the expression of drug-resistant genes in tumor cells [254,255]. Therefore, cancer patients may acquire resistance to anti-PD-1 therapy because of the interaction between TAMs and TME. This discovery lends credence to integrating anti-PD-1/PD-L1 treatment with macrophage targeting.

### 6.3. Macrophage Immune Responses to Anti-PD-1/PD-L1 Therapy

According to prior investigations, the TME is affected by PD-1 inhibitors in several tumors [256]. Researchers used scRNA-seq to demonstrate that the TME in NSCLC patients treated with neoadjuvant PD-1 blockade and chemotherapy was transformed. TAMS shifted from an anti-tumor phenotype to a neutral one [257]. In addition, anti-PD-L1 treatment may inhibit tumor growth by lowering PD-L1 expression and raising the levels of two co-stimulatory molecules, CD86 and MHC-II [258]. Furthermore, anti-PD-L1 treatment improved macrophage phagocytic capacity and immunological function, activating T cells in the TME and eliminating cancer cells [258]. Hence, in specific individuals, anti-PD-L1 treatment has the potential to repolarize macrophages, improve their phagocytic capacity, and alleviate TME.

### 6.4. Clinical Efficacy of Anti-PD-1/PD-L1 Therapy

Current clinical studies highlight significant challenges in translating the preclinical success of anti-PD-1 therapies targeting TAMs into improved patient outcomes. A key limitation involves TAMs’ role in sequestering anti-PD-1 antibodies via Fcγ receptor (FcγR)-mediated capture, rapidly removing these drugs from T cells and diminishing their efficacy in vivo [259]. This mechanism is compounded by the immunosuppressive TME, where PD-1-positive TAMs correlate with reduced survival in cancers like bladder carcinoma and HCC. This often exhibits M2-like polarization that suppresses T cell activity through PD-L1 expression, IL-10 secretion, and Fas ligand interactions [134,260]. Clinical trials combining anti-PD-1/PD-L1 agents with a TGFβ inhibition strategy, successful in preclinical models, have shown limited efficacy, with one phase 1b trial reporting only a 3.1% partial response rate in metastatic pancreatic cancer [261]. Additionally, the paradoxical role of PD-L1-expressing TAMs, which may be associated with better survival in non-immunotherapy contexts but contribute to resistance during treatment, complicates therapeutic targeting [262]. Efforts to overcome these barriers include FcγR blockade to prolong anti-PD-1 engagement and macrophage repolarization strategies, although human trials lag behind preclinical successes [134]. These findings underscore the need for refined biomarkers and combination approaches addressing TAM plasticity and bidirectional interactions with checkpoint inhibitors.

## 7. Targeting TAM Immunotherapy

Building on advances in immunotherapy that leverage macrophages and anti-PD-1/PD-L1 checkpoint inhibitors, an emerging focus has been the development of strategies specifically targeting TAMs to overcome resistance and enhance the efficacy of these treatments [134,252,263]. Future research into TAM might provide fruitful results due to its role in tumor immunity and progression. There are now two main approaches to treating macrophages: reducing the amount of TAM or reprogramming it. Combinations of immunotherapy, chemotherapy, and radiation were prevalent in clinical trials aiming to increase the effectiveness of treatment by targeting TAMs (Table 2) [264,265,266,267,268,269,270,271,272,273,274,275].

### 7.1. TAMs in Clinical Therapy

There has been much discussion in the field of tumor therapy about clinical studies that target TAMs. Various therapeutic techniques have been developed to target TAMs, prevent their formation and aggregation, and control their polarization. Understanding the role of TAMs in the TME, these approaches aim to improve anti-tumor immune responses by modifying their function. Current clinical trials focus on investigating new immunotherapies and therapeutic drugs that target TAMs [90].

#### 7.1.1. Chemokine Inhibitor

Carlumab showed a manageable safety profile, but it failed to demonstrate anti-numerable activity as a monotherapy in this refractory prostate cancer population. The trial highlighted challenges in targeting the CCL2/CCR2 axis alone for TAM modulation in advanced malignancies [276]. BMS-813160, a dual CCR2/CCR5 antagonist, was investigated with nivolumab for advanced renal cell carcinoma to change TAMs and improve anti-PD-1 efficacy in a phase 2 clinical trial, NCT02996110. Preclinical and parallel research suggests CCR2/5 inhibition may lessen immunosuppressive myeloid cell infiltration and synergize with checkpoint inhibitors. This and other trials’ efficacy and safety data are crucial for TAM-targeting approaches like BMS-813160, which involve compensatory mechanisms and varying patient responses [277]. The clinical trial NCT03184870 examined BMS-813160’s capacity to prevent immunosuppressive cell migration and synergize with checkpoint inhibitors. Pending efficacy and safety data will determine whether this TAM-targeting method warrants larger phase 3 trials or immunotherapy combinations [278]. By targeting TAMs to boost anti-tumor immunity, BMS-813160 showed promise in another phase II trial (NCT04123379). The trial’s completion indicates TAM recruitment modulation to improve checkpoint inhibitor responses. Phase III confirmation of this combo therapy is crucial, especially for HCC patients with robust TAM infiltration, where CCR2/5 inhibition and PD-1 blocking could overcome immunosuppressive microenvironments [90]. Patients in the pancreatic cancer phase 1b trial (NCT01413022) with PF-04136309, a CCR2 inhibitor, plus FOLFIRINOX showed acceptable safety, a 49% objective response rate, and fewer TAMs. It changed the TME, although the regimen exhibited greater anti-numerable effectiveness than FOLFIRINOX alone. Likewise, grade ≥ 3 adverse events, such as neutropenia (69%) and febrile neutropenia (18%), prompted safety concerns necessitating careful management. These preliminary findings suggest future research on efficacy and biomarker-driven strategies to expand CCR2 inhibition’s therapeutic window in immunosuppressive TAM populations [265].

#### 7.1.2. CSF1R Inhibitor

Despite low objective response rates, the phase I/II trial (NCT02584647) of pexidartinib (PLX3397) and sirolimus showed acceptable safety and tolerability and prolonged tumor stabilization in some patients with MPNST and TGCT. M2 macrophage infiltration and mTOR pathway suppression were lower in the correlative analyses. However, CSF1R inhibition and immunotherapies may improve efficacy, especially in MPNST subtypes with prior immunological activity [279]. PLX3397, a CSF1R inhibitor, and pembrolizumab in phase 1/2a (NCT02452424) showed safety, and a 1600 mg/day phase 2 dose was used. In this trial, melanoma and gastrointestinal cancers are explored in detail. Preclinical evidence supported targeting TAMs to improve anti-PD-1 efficacy. However, varying clinical outcomes across tumor types underlined the need for biomarker-driven research to identify responsive populations and refine combination methods [280]. The NCT01596751 glioblastoma trial showed good tolerability and blood–tumor barrier penetration for PLX3397, although 8.6% 6-month progression-free survival and no objective responses were achieved. Although monotherapy for glioblastoma has limitations, PLX3397’s ability to modulate TAMs and immune microenvironments is combined with chemotherapy or immunotherapy and biomarkers to identify responsive patients [281]. In recurrent glioblastoma, PLX3397’s phase II trial (NCT01790503) revealed safe blood–tumor barrier penetration and reduced TAMs, but progression-free survival did not increase. This shows that single-agent efficacy is limited. PLX3397 modifies TAMs and boosts CD8^+^ T cell infiltration in sarcomas, suggesting it may work with immunotherapies in specific cancers [281]. In phase I/II trials (NCT02829723), BLZ945 showed favorable safety, pharmacokinetics, and early immune modulation in the TME. However, monotherapy efficacy was restricted in advanced solid tumors. CSF1R reduction in TAMs with radiation or checkpoint inhibitors repolarizes TAMs and activates cytotoxic T cells, improving longevity in preclinical trials [282].

#### 7.1.3. Antibody-Targeting CSF1R

In clinical studies, LY3022855, a CSF-1R inhibitor, reduced TAMs and raised pro-inflammatory cytokines, but it had a limited anti-cancer effect, with some patients having stable illnesses. Combining LY3022855 with bevacizumab and radiation in NCT03101254’s glioblastoma research showed that TAM-mediated immunosuppression can be overcome. Moreover, monotherapy activity was minimal [283]. In another trial (NCT02923739), Emactuzumab lowered TAMs by inhibiting CSF1R, improving anti-cancer immune responses and objective response rates up to 86% in diffuse-type tenosynovial giant cell tumors in early-phase trials. The medication improved patient-reported outcomes and exhibited lasting responses. However, grade 3 facial edema and autoimmune-like reactions require phase 3 trials to validate long-term benefits and optimize safety [284]. Emactuzumab, a CSF1R inhibitor, improved patient-reported outcomes in a trial (NCT03193190) in diffuse-type tenosynovial giant cell tumors (dt-GCT) with durable responses (71% objective response rate) and fewer TAMs. However, only 7% of solid tumors react to treatment with paclitaxel, highlighting the need for targeted applications and combination therapies to boost therapeutic potential [268]. AMG 820, a CSF1R inhibitor, combined with pembrolizumab in the phase Ib/II trial (NCT02713529) showed acceptable safety and pharmacodynamic evidence of target engagement (e.g., increased serum CSF1/IL-34, reduced CD16^+^ monocytes). Still, limited anti-imagined activity was observed, with only 3% of patients showing partial responses. The preclinical rationale supported CSF1R inhibition to regulate TAMs and improve checkpoint inhibitor efficacy. At the same time, clinical data demonstrated biomarker-driven patient selection or other combination approaches to boost therapeutic potential [271]. The phase 1b/2 trial (NCT02880371) combining ARRY-382 (a CSF1R inhibitor) and pembrolizumab showed tolerable safety but limited clinical efficacy. The trial showed partial responses in only 3 out of 76 patients (10.5% in phase 1b and 3.7% in a phase 2 pancreatic cancer cohort) and no further development. The preclinical rationale supported targeting TAMs to boost anti-PD-1 activity, but trial results suggest CSF1R inhibition alone may not overcome immune resistance in advanced solid tumors. Optimizing combinations of biomarker-selected populations is biologically plausible [272].

#### 7.1.4. Anti-CD40 Agonist

In a clinical trial (NCT03214250), APX005M with CSF1R inhibition exhibited tolerable safety and pharmacodynamic immune modulation, including pro-inflammatory cytokine overexpression and macrophage reprogramming. Nevertheless, there was only a 4% partial response and 31% stable disease in PD-1/PD-L1-resistant tumors. These preliminary findings imply that immunosuppressive TAMs can be overcome, although dose regimens and checkpoint inhibitor combinations may be needed to transform innate immune activation into potent anti-cancer responses [285]. Moreover, preclinical studies reveal that CSF1R inhibitors and CD40 agonists like APX005M reprogram TAMs to overcome pancreatic cancer immunosuppression and improve T cell priming and chemosensitization. Early-phase trials like NCT03214250 (APX005M + chemotherapy ± nivolumab) demonstrated safety and pharmacodynamic activity, but limited monotherapy efficacy suggests the need to optimize dosage regimens and biomarker-driven patient selection for future combinations [286].

### 7.2. Clinical Trial Simulation and TAM Physiology Integration for Cancer Therapy

Current innovations in cancer research highlight the critical integration of TAM physiology with computational modeling to optimize therapeutic strategies and clinical trial design. Depending on their polarization state, TAMs exhibit dual pro- and anti-tumor roles and are increasingly targeted through reprogramming agents to counteract immunosuppression and enhance treatment efficacy [52,287]. For example, computational frameworks like Boolean dynamical networks have successfully identified pathways driving TAM pro-metastatic activity in breast cancer. This enables predictions of pharmacological interventions that reduce metastasis in patient-derived models [288,289]. Similarly, quantitative systems of pharmacology models simulating virtual metastatic triple-negative breast cancer patients have improved immunotherapy outcome predictions by analyzing on-treatment biomarkers, such as early tumor diameter changes [289]. These approaches address challenges like tumor heterogeneity and therapy resistance by combining TAM-targeted strategies (e.g., CSF-1R inhibition, PI3K pathway modulation) with immune checkpoint inhibitors or chemotherapy [52,82,290]. However, clinical translation requires more profound insights into TAM plasticity, spatial distribution, and patient-specific biomarker profiles to refine combination therapies and trial simulations [52,287,289].

Moreover, cancer immunotherapy TAMs are better understood by utilizing quantitative systems pharmacology (QSP) modeling. Immunosuppression and treatment response macrophage dynamics have been simulated using mechanistic QSP platforms. CCR2 inhibitors and anti-PD-L1 may boost efficacy in M1/M2 macrophage polarization models and recruitment via CCL2-CCR2 signaling, especially in individuals with substantial TAM infiltration or checkpoint inhibitor resistance [289,291]. TGF-β-mediated immunosuppression may lead to poorer treatment response in virtual patients with higher M2-like macrophage density. Integrating data into modular QSP frameworks allows for macrophage-targeted therapy predictions and population-level validation versus clinical trial findings. This computational technique optimizes combination medications and finds patient subgroups that may benefit from TAM activity modulation [289,291]. Future directions focus on refining spatial and multi-scale models to capture TME complexity, optimizing combination therapies, and personalizing treatment based on TAM profiles [291,292]. Additionally, expanding QSP frameworks to include novel macrophage subtypes and their interactions with stromal components could enhance predictive accuracy for clinical translation [291,292].

### 7.3. Reduced TAM Levels

One potential cancer treatment technique, whether used alone or in conjunction with chemotherapy, is the depletion of macrophages in the TME. When the signal transduction axis inhibits the CSF1/CSF1R receptor, macrophages undergo apoptosis. This axis is critical for the survival of macrophages. To begin with, T cell responses may be enhanced by blocking CSF-1R in conjunction with radiation or chemotherapy. Glioblastoma brain tumors may have their survival time extended by stimulating the CD8^+^ T cell response and depleting the immunosuppressive TAM by blocking CSF1R signaling [282]. Some cancers, including orthotopic glioblastoma and localized prostate cancer, are now undergoing clinical studies that combine chemotherapy with CSF1R inhibitors [282,293]. Moreover, specific immunotherapies, such as CD-40 agonists [294] and PD-1 inhibitors [295], may be more effective by inhibiting CSF1/CSF1R.

Another way to decrease TAMs in the TME is to prevent monocyte recruitment from circulating to the tumor site, as TAMs are derived from monocytes. Transduction of signals between C-C motif ligand 2 (CCL2) and CC chemokine receptor 2 (CCR2) is essential for the recruitment of monocytes from bone marrow to the location of tumors [296]. When CCR2 is inhibited, monocytes stay in the bone marrow instead of circulating in the bloodstream. This decreases the number of TAMs, which reduces the recruitment of monocytes to the tumor site and metastatic foci, reducing the tumor and improving survival [297,298,299].

Macrophage recruitment also involves the CXCL12-CXCR4 and the angiopoietin 2 (ANG2)–TIE2 axis, among others [300,301,302]. Consequently, TEM depletion might lead to vascular deterioration, ANG2 neutralization could enhance the response to vascular VEGFA blockage, and TEM recruitment inhibition could decrease tumor development [303].

#### TAM Elimination with CAR-T Cells

Improving anti-tumor immunity using CAR T cell treatment that targets TAMs is a new approach to overcoming immunosuppressive TME. Reprogramming the TME to promote endogenous immune responses and enhance the effectiveness of traditional immunotherapies is possible via the selective depletion of immunosuppressive TAM subsets by CAR-T cells [304].

FRβ-specific CAR-T cells have demonstrated success in preclinical models. Depletion of folate receptor β (FRβ)-expressing TAMs reduces immunosuppressive M2-like macrophages, enriches pro-inflammatory monocytes, and recruits tumor-specific CD8^+^ T cells [305,306]. This approach delays tumor progression and prolongs survival in syngeneic mouse models (e.g., OC) without directly targeting tumor cells [307]. FRβ+ TAMs suppress CAR-T and endogenous T cell activity by inhibiting proliferation and IFN-γ secretion [308]. Other targets under investigation include FAP (fibroblast activation protein) on CAFs and CD123 on immunosuppressive myeloid cells [307].

CAR-T-mediated TAM elimination induces systemic immune remodeling. In myeloid cells, reduced IL-10, TGF-β, and VEGF secretion diminishes immunosuppression and angiogenesis [309]. In endogenous T cell activation, depleting FRβ+ TAMs increases MHC-II expression on remaining macrophages, enhancing antigen presentation and stimulating tumor-specific CD8^+^ T cells [305]. Moreover, CAR-T-derived IFN-γ and GM-CSF promote a pro-inflammatory TME6 in cytokine modulation.

In therapeutic responses, sequential administration of CAR-T products improves outcomes. Preconditioning with FRβ CAR-T cells before tumor-directed therapies (e.g., anti-mesothelin CAR-T) enhances engraftment and efficacy [310,311]. Simultaneous co-administration of TAM-targeting and tumor-targeting CAR-Ts is less effective, suggesting that timed depletion of immunosuppressive cells is critical [305,312]. There are several challenges and innovations. Within tumor heterogeneity, non-uniform TAM antigen expression may limit efficacy [307]. Suicide gene systems (e.g., inducible caspase-9) are being explored to mitigate off-tumor toxicity [313]. Concerning synergy with lymphodepletion, combining CAR-T therapy with chemotherapy amplifies anti-tumor responses [305,314]. This approach highlights the potential of CAR-T cells as “TME-conditioning agents” rather than direct tumor killers, offering a pathway to enhance adoptive immunotherapies in solid tumors.

### 7.4. TAM Reprogramming

After TAMs are eliminated, the immune-stimulating role of macrophages declines, as they are the primary phagocytes and antigen-presenting cells in the tumor. Therefore, a better way to treat cancer would be to repolarize or reprogram the TAM such that it has more anti-tumor functions and fewer tumor-promoting effects. For instance, in the breast cancer mouse model, blocking IL-10 signal transduction greatly enhances chemotherapy effectiveness because the TAM is the primary source of IL-10. To counteract the anti-tumor effect of paclitaxel and carboplatin on CD8^+^ T cells, a TAM secretes IL-10, suppressing the production of IL-12 by APCs [72]. Furthermore, TAM repolarization causes it to produce the pro-inflammatory cytokine IFN-α in a tumor setting, potentially activating NK and T cells. This considerably reduces tumor development in a mouse model [315]. Inhibiting histone deacetylase (HDAC) may reprogram macrophage epigenetics and activate T cells for an immunological response [316,317]. A specific class, IIa HDAC inhibitor, increased the efficacy of chemotherapy and immune checkpoint inhibitors in a breast cancer model by inducing an anti-tumor macrophage phenotype and enhancing the T cell immunological response [316]. Furthermore, macrophage immunosuppression in TAM may be driven by PI3K signaling activation, while T cell responses can be improved by reprogramming macrophages to decrease PI3K signaling [318,319].

### 7.5. Therapy Using CAR–Macrophages

In hematological cancers, CAR-T cells are beneficial, but, in solid tumors, where T cell infiltration is limited, the effectiveness of CAR-T treatment is still limited [320,321]. Nevertheless, this drawback is circumvented by CAR–macrophages (CAR-M), as circulating monocytes may resupply the macrophages in the TME. Macrophage polarization, anti-cancer activity, higher phagocytic capacity, and antigen-dependent activities, including the production of cytokines, might be improved by CAR expression [322]. CAR-M cells have anti-tumor effects in both primary and metastatic cancers, support phagocytosis, and display M1 capabilities somewhat consistently [323]. The anti-cancer effectiveness of CAR-M in various tumor types is now being investigated in many ongoing or planned clinical studies.

### 7.6. Integrating Anti-PD-1 Treatment with Macrophages Targeting Cancer

In vivo and in vitro studies on cancer treatment combining macrophages targeting with anti-PD-1 therapy have been conducted [258,324,325,326]. In hepatocellular carcinoma, repolarization of TAM might enhance the efficiency of anti-PD-1 therapy, as mentioned before, and is a potentially helpful technique for cancer treatment [326]. Radiotherapy and chemotherapy can potentially improve the effectiveness of immunotherapy for cancer by resetting macrophages to an M1 phenotype [325]. To improve the survival result of tumor-based immunotherapy, vinblastine may activate NF-κB, increase CD8^+^ T cell populations, and polarize TAMs to the M1 phenotype [325]. By reprogramming immunosuppressive TAMs, bi-target treatments like PD-1-IL-2 cytokine variation (IL2v) boost therapeutic effectiveness [324]. IL2v uses anti-PD-1 as a target moiety fused with an immuno-stimulatory IL2v. Ultimately, it seems that a potential approach to combating medication resistance in cancer patients might include combining anti-PD-1 treatment with macrophage targeting.

## 8. Contradictory Results Regarding TAM Depletion Versus Reprogramming Strategies

The contradictory results between TAM depletion and reprogramming strategies stem from the complex biology of TAMs, TME heterogeneity, and the dual roles macrophages can play in cancer progression. Studies of TAM depletion strategies with limited efficacy in most cancers, including CSF1/CSF1R inhibitors (e.g., PLX3397), showed limited monotherapy efficacy, except in CSF1R-driven tumors, like diffuse-type giant cell tumors [52]. The efficacy of TAM depletion hinges on tumor biology, microenvironmental adaptations, and therapeutic context. While CSF1R inhibitors excel in CSF1R-dependent tumors like TGCT, most cancers require multimodal strategies to overcome resistance. Future efforts should focus on subset-specific TAM targeting, tumor-immune spatial mapping, and rational combinations to unlock the full potential of macrophage-directed therapies [52,327]. 

Systemic depletion risks disrupting homeostatic macrophage functions, increasing infection susceptibility and inflammation [328]. TAM depletion enhanced anti-PD-1 therapy in postoperative settings by reducing immunosuppressive signals and improving T cell infiltration [329]. However, excessive depletion in immunocompromised patients worsened outcomes [60]. The contradiction arises from non-selective depletion strategies. Precision approaches, such as limited delivery, M2-specific targeting, and combination therapies, enhanced anti-PD-1 efficacy while preserving immune homeostasis. Patient-specific factors (immune status, tumor type) and dynamic monitoring are critical to balancing benefits and risks [329].

In TAM reprogramming strategies, shifting M2-like to M1-like phenotypes reprograms TAMs toward pro-inflammatory M1-like states via CD40 agonists, TLR activators, or radiotherapy, which showed anti-tumor effects in preclinical models [330]. This approach preserves macrophage-mediated tissue repair while reducing immunosuppression [5]. Similarly, these highlight the importance of multimodal approaches targeting macrophage plasticity and the broader TME to sustain anti-tumor immunity. To avoid T cell exhaustion, future research should concentrate on reducing M2 TAMs (via CSF-1R inhibitors) before initiating M1 reprogramming, combining with CD40/TLR agonists, and monitoring TAM polarization using miR-7083-5p or CSF2RA levels [331,332].

Clinical translation has several challenges, such as overlapping therapeutic targets (e.g., CCR2, CD47), leading to redundant clinical trials with mixed results [52]. Similarly, TAM plasticity reverses pro-tumor states post-treatment, limiting durability [61]. While no single approach fully resolves these contradictions, integrating mechanistic insights with engineered therapies and adaptive trial designs offers a path toward durable responses. For example, combining 47E-modified CAR T cells with timed CD47 blockades and TAM-reprogramming agents could simultaneously enhance T cell persistence, myeloid activation, and plasticity suppression [333,334]. Table 3 shows vital contradictory factors.

### Future Therapeutic Perspectives

Ineffectiveness hinders clinical medication development the most. In clinical studies, monotherapy therapies showed modest response rates (~5%). Biomarkers, cancer type, stage, and therapy optimization may raise them. TAM subgroups must determine whether a cancer type requires one therapy or several TAM-reprogramming drugs. Future studies will show whether patient selection improves monotherapy [86,90]. Drugs that target the same targets increase patient treatment and technology development expenses. Precise treatment comparisons and research are best performed early. Patient ex vivo tumor models may predict therapy-induced anti-tumor immunity [335]. Many pro-inflammatory mediators are context-dependent and may not predict treatment outcome, so parameters should be picked carefully. IL-1β and HMGB1 may contribute to cancer growth and progression [52]. Although essential for anti-tumor immune responses, persistent and intense IFN-γ signaling may increase cancer growth [89]. TME immune responses are complicated. Thus, whole systems, cytokines, must be investigated.

Macrophage-specific therapy and TAM biology should be investigated after therapy. Trials show that suitable treatment targets can elicit proper immune activation without activating counteractive pathways or harmful feedback mechanisms, alter TAMs rather than healthy tissue macrophages, lack TME compensatory pathways, and therapeutically manipulate solid tumors [52,90]. Healthy tissue macrophages may lose therapeutic benefits due to cancer-specific expression, persistent inflammation, or monocyte-derived macrophages. Side effects restrict the treatment window. However, TME-specific targets or delivery channels may improve specificity [335,336]. Transforming monocytes into TAMs may facilitate tissue-free solid tumor invasion.

Understanding TAM biology improves treatment. The scRNA-seq shows TAM subgroup complexity that clinically proven drugs cannot treat. TAM biology is too complex for preclinical orthotopic mouse tumor models [337,338]. Recently found TAM states may have anti- and pro-tumoral effects in the same cell subsets, making their removal difficult. Managing anti-tumoral functions may require subset-specific regulation. Flexible macrophages can conduct TAMs, although intratumoral localization is required [82,87,339]. TME limits TAM responses. Thus, therapy must overcome this to modify behavior. Treatment effects on subsets and interactions to reprogram TAMs should be studied.

Similarly, recent advances in TAM research highlight the need for standardized classification systems and biomarker discovery pipelines to improve clinical translation. Emerging strategies emphasize integrating scRNA-seq with spatial transcriptomics and mIHC to resolve TAM heterogeneity while preserving spatial context [51]. Combining bulk RNA-seq with scRNA-seq data for biomarker development enables the identification of prognosis-linked TAM subsets and their gene signatures, as demonstrated in lung and colorectal cancers [41]. Standardization efforts focus on universal TAM quantification protocols, such as fixed tissue section thickness and consensus “hot spot” field counts. It further enhances reproducibility across studies [340]. The CyTOF and AI-driven image analysis platforms are being leveraged to decode TAM functional states and surface protein profiles, paving the way for context-specific biomarkers [341]. Future directions also prioritize multi-omics deconvolution to disentangle TAM ontogeny, plasticity, and tumor-niche interactions, which could yield targeted immunotherapy strategies [8].

Finally, TAM-targeted medications must be combined, although TAMs affect other cancer therapies and progression. Many studies have shown that chemotherapy, radiation, and immunotherapy work [86,290]. Anti-inflammatory TAM polarization and immunosuppression, or pro-inflammatory TAM polarization and cancer cell death, may result from drug and radiation dosages [52,330]. TABectedin reveals that decreasing monocytes or TAMs may boost chemotherapy’s anti-tumor effects [52]. Many monoclonal antibodies exploit macrophage-mediated effector action and anti-tumor immunity. However, TAM-regulated immunosuppression and T cell exclusion restrict immunotherapy. Combination medicines need data-driven and mechanism-based approaches [52,82,342,343]. Eliminating TAM-mediated T cell suppression may sensitize tumors to immune checkpoint inhibitors (ICIs). Several therapies may treat TAM and TME deficiencies [344,345,346]. CSF-R1 inhibition enhances IGF-1 and PI3K signaling in murine glioma, creating treatment resistance [336]. PI3K inhibitors may help. Researchers must address these difficulties and improve treatment regimens to optimize the potential of TAM-reprogramming drugs [347]. Although challenging to target, TAMs are vital to the TME and should not be overlooked to improve cancer treatment.

## 9. Conclusions

TAMs play a multifaceted role in cancer progression, shaped by their origin, polarization, and dynamic crosstalk within the TME. Derived from circulating monocytes or tissue-resident precursors, TAMs exhibit plasticity between pro-inflammatory M1 and immunosuppressive M2 phenotypes, driven by cytokines like IFN-γ/LPS (M1) or IL-4/IL-13 (M2), with M2 polarization often dominating in advanced tumors [5,67]. Recent advances in scRNA-seq, mass cytometry, and spatial transcriptomics have revealed extensive heterogeneity in TAM subsets, enabling precise characterization of their functional states and spatial distribution within tumors [57,347]. Within the TME, TAMs accelerate tumorigenesis by promoting angiogenesis through VEGF, extracellular matrix remodeling via MMPs, EMT, and immune suppression through PD-L1 expression and IL-10 secretion [63,102,251]. Bidirectional signaling between TAMs and tumor cells, such as CSF-1/EGF loops, foster a pro-metastatic niche, while TAM-derived factors like TGF-β and PGE2 enhance cancer stem cell plasticity [100,102]. Immunotherapies targeting TAMs include strategies to block recruitment (anti-CCL2/CCR2), repolarize M2 to M1 (PI3Kγ inhibitors), or enhance phagocytosis (anti-CD47), often combined with PD-1/PD-L1 checkpoint inhibitors to counteract TAM-mediated immunosuppression [134,347,348]. However, therapeutic approaches face contradictions; while TAM depletion (e.g., CSF-1R inhibitors) reduces tumor burden, it may impair anti-tumor immunity, whereas reprogramming strategies risk incomplete phenotype conversion or context-dependent efficacy [347,348]. These complexities underscore the need for tailored, multimodal therapies integrating TAM targeting with broader immune modulation to address macrophages’ dualistic roles in cancer progression and treatment resistance.

## Figures and Tables

**Figure 1 cells-14-00741-f001:**
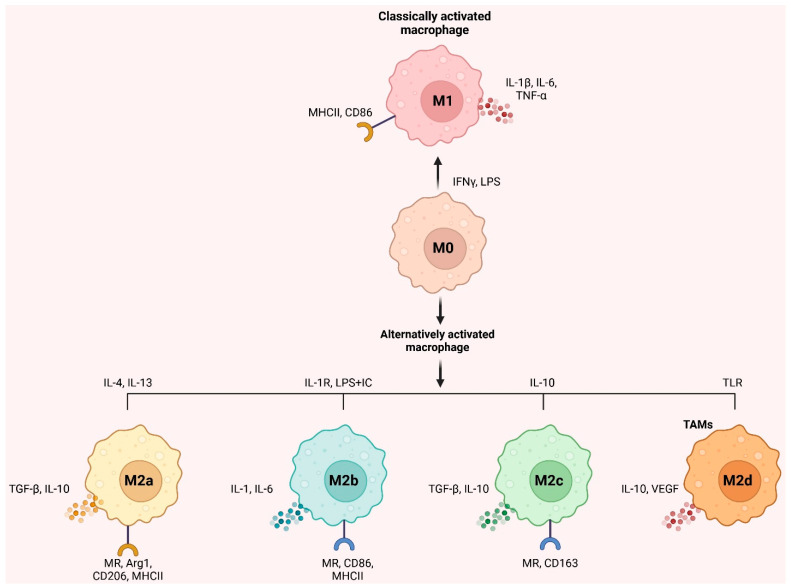
How various environmental stimuli influence the trajectory of macrophage differentiation. Various interferons, LPS, and interleukins polarize classically activated macrophages into alternatively activated macrophages. Created in BioRender. Saeed, A. (2025) https://BioRender.com/ient5ad.

**Figure 2 cells-14-00741-f002:**
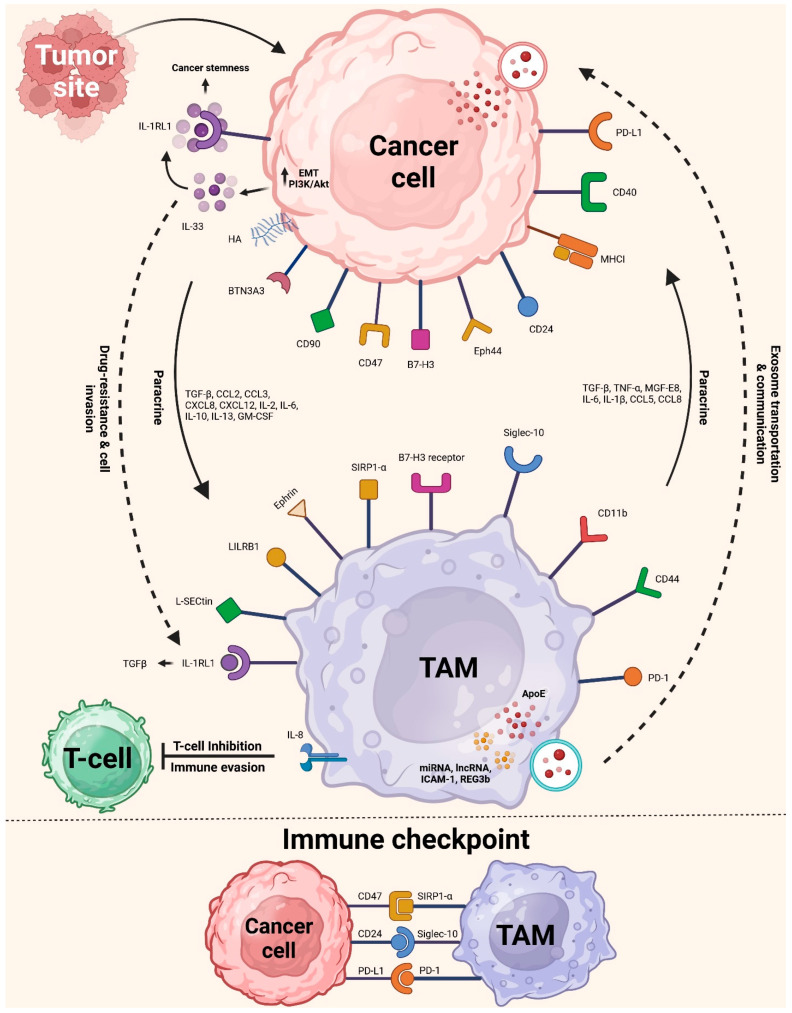
TAMs produce cancer-promoting IL-6, IL-8, and IL-10. Similarly, they produce T-cell-killing IL-8. Multiple tumor cell–TAM juxtacrine connections restrain immunity. PD-1/L1 inhibits macrophages to hide malignancy from the immune system, and, through B7-H3, cancer immune evasion suppresses T cells. The “do-not-eat-me” signal is sent via SIRPα/CD47 and CD24/Siglec-10 pathways. Tumor cells may overexpress CD47 and CD24 to escape macrophages. LILRB1/MHC class I component β2-microglobulin is essential for tumor defense against macrophages. Cells exchange circRNA, miRNA, lncRNA, ICAM-1, REG3b, lipids, proteins, and mRNA via exosomes. TAM associate ApoE and other exosome-delivered chemicals activate PI3K-Akt, causing epithelial–mesenchymal transition (EMT), cytoskeletal remodeling, and cancer cell migration. Eph44-Ephrin juxtacrine controls immune cell motility, activation, survival, and death. CD44 affects phagocytosis, inflammation, and multidrug resistance and is an effective phagocytic receptor due to HA binding and metabolization. Eph44-ephrin, CD90-CD11b, BTN3N3-L-SECtin, and CD44-HA all have a role in keeping cancer stem cells alive. The paracrine system promotes TAM proliferation, differentiation, and TGF-β production, leading to tumor cell invasion and treatment resistance. The autocrine relationship between IL-1RL1 and IL-33 preserves the tumor cell’s stemness. Thus, the complex TME interactions promote crosstalk. ApoE, Apolipoprotein E; CCL, C-C motif chemokine ligand; EMT, epithelial–mesenchymal transition; GM-CSF, granulocyte–macrophage colony-stimulating factor; HA, hyaluronic acid; IL, interleukin; JAK, Janus kinase; lncRNA, long non-coding RNA; M-CSF, macrophage colony-stimulating factor; MGF-E8, milk fat globule–EGF factor 8; MHC, major histocompatibility complex; miRNA, microRNA; PD-L1, programmed death-ligand 1; Siglec, sialic acid binding Ig-like lectin. Created in BioRender. Saeed, A. (2025) https://BioRender.com/fuol6ul.

**Table 1 cells-14-00741-t001:** Metabolic plasticity across different activation states.

Metabolic Features	M1-like TAMs	M2-like TAMs
Primary metabolism	Glycolysis	Mixed glycolysis/OXPHOS
Key pathways	HIF-1α, PKM2	mTORC2, FAO
Functional output	Pro-inflammatory (limited)	Immunosuppression, tissue repair

**Table 2 cells-14-00741-t002:** Clinical studies of several drugs that target TAMs.

Drug	Phase	Cancer Type	Combination Therapy	NCT Identifier
**Chemokine inhibitors**				
Carlumab (anti-CCL2 antibodies; Centocor)	Phase II (completed)	Prostate cancer	NA	NCT00992186
BMS-813160 (CCR2/CCR5 antagonist; Bristol Myers Squibb)	Phase II (completed)	Renal cell carcinoma	Nivolumab (OPDIVO) in conjunction with ipilimumab (Yervoy)	NCT02996110
	Phase I/II (completed)	Pancreatic cancer, colorectal cancer, non-small cell lung cancer	Nab-paclitaxel with nivolumab	NCT03184870
	Phase II (ongoing)	Hepatocellular carcinoma	Nivolumab	NCT04123379
PF-4136309 (CCR2 antagonist; Pfizer)	Phase II (completed)	PDAC	Nab-paclitaxel, gemcitabine	NCT01413022
**CSF1R inhibitors**				
PLX3397 (Plexxikon)	Phase I/II (ongoing)	Tumors of the nerve sheath and sarcoma	Sirolimus (Rapamune)	NCT02584647
	Phase I/II (terminated)	Melanoma and solid tumors, both at advanced stages	Pembrolizumab (Keytruda)	NCT02452424
	Phase I/II (completed)	Breast carcinoma	Eribulin (Halaven)	NCT01596751
	Phase I/II (completed)	Glioblastoma	Radiotherapy, temozolomide (TMZ)	NCT01790503
BLZ945 (Novartis)	Phase I/II (terminated)	Solid tumors	PDR001 (anti-PD1)	NCT02829723
**Antibodies targeting CSF1R**				
LY3022855 (Eli Lilly’s IMC-C S4)	Phase I/II (completed)	Melanoma	MEK/BRAF inhibitors	NCT03101254
Emactuzumab (RO5509554/RG7155; Roche)	Phase II (terminated)	Gynecological neoplasms and OC	Gynecological neoplasms and OC	NCT02923739
	Phase I/II (ongoing)	PDAC	Nab-paclitaxel, gemcitabine	NCT03193190
AMG820 (Amgen)	Phase I/II (completed)	Pancreatic cancer, CRC, NSCLC	Pembrolizumab	NCT02713529
ARRAY-382 (Pfizer)	Phase I/II (terminated)	Solid tumors	Solid tumors	NCT02880371
**Agonist anti-CD40 antibodies (cont.)**				
APX005M (Apexigen)	Phase II (completed)	Esophageal cancer	Radiation, paclitaxel, carboplatin	NCT03214250
	Phase I/II (completed)	Pancreatic cancer	Nab-paclitaxel, gemcitabine, nivolumab	NCT03214250

**Table 3 cells-14-00741-t003:** Key factors explaining contradictions.

Factor	Impact on Depletion vs. Reprogramming	Ref
Tumor Type	CSF1R inhibitors succeeded in CSF1R-addicted tumors but failed in others (e.g., pancreatic cancer).	[52]
TME Heterogeneity	TAM subsets exhibit divergent functions (e.g., angiogenesis vs. immunosuppression), complicating targeting.	[61]
Therapeutic Timing	Post-surgery was depleted to prevent recurrence, while reprogramming was synergized with immunotherapy.	[86,329]
Combination Therapies	Reprogramming + checkpoint inhibitors improved responses in resistant tumors, whereas depletion alone rarely sufficed.	[52]

## Data Availability

No new data were created or analyzed in this study.

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
