# Peer review of "Tumor-Associated Macrophages: Polarization, Immunoregulation, and Immunotherapy"

_cells, 2025, doi:10.3390/cells14100741_

Round 1
Reviewer 1 Report
Comments and Suggestions for Authors
The manuscript comprehensively reviewed the key findings and questions as well as the future research directions about tumor-associated macrophages, with a focus on their activation states, immunoregulation mechanisms, and immunotherapeutic potentials. It is well-organized and informative and the references are current.
Some general comments are:
- Since there are review papers on TAMs, it would be very helpful to state how this one differs from others, how this one addresses the current knowledge gaps in this field that have not been specifically reviewed by others.
- It would be helpful to describe the strategies that the author has adopted to cover the literature on this topic.
- Two aspects seem to be left out. One is the differences in TAMs between solid and liquid tumors. The other is metabolism that supports TAMs, esp. metabolic characteristics of TAMs in different activation states. Was there special consideration that these were not included in the manuscript?
- Since Car T therapy has very limited success in treating solid tumors with TAMs being a significant contributor to treatment resistance, research on TAMs elimination with Car T cells is worthy of reviewing.
Specific comments:
- Line 12, “anti-PD-L-1”, should it be “anti-PD-L1”?
- Line 39, “and” after TAMs should be removed.
- Line 83, “after they arrive”, it would be beneficial to mention the timelines for their various function in tumor cell migration, metastasis, and tumor growth.
- Line 146, for citation [55], the first author should be mentioned not the senior author, alternatively, it could be “Weissman group”, instead Weissman et al.
- Line 149, please double check this number 70%. The survival curve Fig. 5b does not seem to indicate 70% higher.
- Line 154-155, citation [59] is about glioblastoma, it might be better to specify it.
Line 196-197, the wording is confusing. It was just stated that M1-like TAMs enhance anti-tumor immunological responses (line 195), but then it is stated “M1-like TAMs attract regulatory T cells and other immunosuppressive cells to help construct an immunosuppressive TME. Do you mean M1-like help construct an immunosuppressive TME? If so, how do they enhance anti-tumor responses?
Author Response
Point-by-point answer to the reviewer's 1 comments
Reviewer 1
Comments and Suggestions for Authors
The manuscript comprehensively reviewed the key findings and questions as well as the future research directions about tumor-associated macrophages, with a focus on their activation states, immunoregulation mechanisms, and immunotherapeutic potentials. It is well-organized and informative and the references are current.
Some general comments are:
Comment 1. Since there are review papers on TAMs, it would be very helpful to state how this one differs from others, how this one addresses the current knowledge gaps in this field that have not been specifically reviewed by others.
Answer. Thank you so much for pointing this out. We agree that there are numerous reviews on TAMs. But here, based on current literature, we detailed macrophage biology in TMEs in this study, stressing their immunosuppressive and tumor development functions and exploring novel therapeutic methods. We also precisely and comprehensively explored TAM cross-talk with other immune cells critical for immunoregulation. Next, it notably incorporates current breakthroughs in TAM characterization technologies like single-cell sequencing and spatial transcriptomics with a crucial review of TAM-targeted therapy contradictions, especially the conflict between depletion and reprogramming. Unlike earlier reviews, it shows how TAMs cause angiogenesis, metastasis, and immune evasion by mapping their bidirectional interaction with cancer cells. It also compares macrophage-centric and anti-PD-1/PD-L1 immunotherapy’s mechanisms. Combining foundational notions (origin, polarization) with translational insights. Likewise, centered on recent advances, this review offers a sophisticated framework for improving TAM-targeted precision cancer therapy.
Comment 2. It would be helpful to describe the strategies that the author has adopted to cover the literature on this topic.
- Two aspects seem to be left out. One is the differences in TAMs between solid and liquid tumors. The other is metabolism that supports TAMs, esp. metabolic characteristics of TAMs in different activation states. Was there special consideration that these were not included in the manuscript?
Answer: Thank you so much for mentioning these critical topics. We agree and have addressed these topics in the revised version of the manuscript. Please see Page 6, Line 230, and Page 7, Line 256.
Also, we have revised the manuscript accordingly to describe the “Search Strategy” to cover the literature for this review. Please see page 2, Line 64.
Comment 3. Since Car T therapy has very limited success in treating solid tumors with TAMs being a significant contributor to treatment resistance, research on TAMs elimination with Car T cells is worthy of reviewing.
Answer: Thank you so much for the valuable comment. We agree and have revised the manuscript accordingly and added research on the elimination of TAMs with Car T cells. Please see Page 24, Line 827.
Specific comments:
- Line 12, “anti-PD-L-1”, should it be “anti-PD-L1”?
Answer: We agree and have revised the typo accordingly. Please see Line 15.
- Line 39, “and” after TAMs should be removed.
- Answer: We agree and have revised accordingly. Please see Line 40.
- Line 83, “after they arrive”, it would be beneficial to mention the timelines for their various function in tumor cell migration, metastasis, and tumor growth.
Answer: Thank you so much for the comment. We agree and have revised accordingly. Please see Line 189.
- Line 146, for citation [55], the first author should be mentioned not the senior author, alternatively, it could be “Weissman group”, instead Weissman et al.
Answer: We agree and have revised accordingly. Please see Line 315.
- Line 149, please double check this number 70%. The survival curve Fig. 5b does not seem to indicate 70% higher.
Answer: We agree and have revised accordingly, please see Line 317.
- Line 154-155, citation [59] is about glioblastoma, it might be better to specify it.
Answer: We agree and have revised accordingly, please see Line 323.
Line 196-197, the wording is confusing. It was just stated that M1-like TAMs enhance anti-tumor immunological responses (line 195), but then it is stated “M1-like TAMs attract regulatory T cells and other immunosuppressive cells to help construct an immunosuppressive TME. Do you mean M1-like help construct an immunosuppressive TME? If so, how do they enhance anti-tumor responses?
Answer: Thank you so much for pointing that out. We agree and have removed the confusing statement from the revised manuscript. Please see Line 362.
The remarks made by the distinguished reviewer on our review are truly appreciated, and we are grateful.

Reviewer 2 Report
Comments and Suggestions for Authors
Your work synthesizes current knowledge regarding the complex roles of TAMs in tumor progression, immunoregulation, and therapeutic applications. After careful evaluation, I offer the following assessment of your manuscript's strengths and areas for improvement.
Strengths
-
Comprehensive Scope: Your review successfully covers multiple dimensions of TAM biology, from origins and polarization to complex interactions within the tumor microenvironment (TME). This breadth provides valuable context for readers across specialties.
-
Clinical Relevance: The manuscript effectively bridges basic immunology with clinical applications, particularly in discussing immune checkpoint inhibitors and emerging immunotherapies. The connection to projected cancer statistics for 2025 establishes immediate relevance.
-
Detailed Cellular Interactions: Your thorough examination of TAM interactions with various immune cells (NK cells, T cells, MDSCs, neutrophils, etc.) highlights the complexity of the TME and provides mechanistic insights into immunosuppression.
-
Visual Representations: The figures appear well-conceptualized to illustrate complex biological processes, particularly the polarization pathways (Figure 1) and TAM-tumor cell interactions (Figure 2).
-
Translational Focus: The emphasis on therapeutic targeting of TAMs (Section 7) demonstrates practical applications of the biological principles discussed earlier, enhancing the review's impact.
Areas for Improvement
-
Critical Analysis of Contradictory Evidence: While you present multiple pathways and mechanisms, the review would be strengthened by more robust discussion of conflicting evidence and unresolved controversies in the field. For example:
-
The dual roles of TAMs in pro/anti-tumor immunity
-
Contradictory results regarding TAM depletion versus reprogramming strategies
-
Inconsistent clinical outcomes from targeting CSF1R pathways
-
-
Balanced Assessment of Clinical Efficacy: The conclusions regarding combination therapies (anti-PD-1 + TAM targeting) would benefit from more critical appraisal of clinical trial limitations, failures, and challenges in translating preclinical success to patient outcomes.
-
Contemporary Context: While the foundation literature is well-represented, incorporating more recent breakthroughs (particularly single-cell RNA sequencing studies post-2020) would strengthen the review's currency and relevance.
-
Quantitative Synthesis: Consider incorporating meta-analytical approaches or quantitative synthesis of prognostic values across studies when discussing the clinical significance of TAM infiltration in various cancer types.
Recommendations for Revision
- Expand discussion of contradictory findings and propose frameworks to reconcile these contradictions
-
Provide more balanced assessment of clinical trial outcomes
-
Incorporate recent technological advances in TAM characterization
-
Consider adding a future directions section that proposes standardized approaches to TAM classification and biomarker discovery
Your manuscript provides valuable insights into TAM biology and therapeutic potential, addressing an important area in cancer immunology. With the suggested revisions, it would make an even stronger contribution to the field by enhancing methodological transparency and providing more nuanced analysis of complex, sometimes contradictory evidence.
Comments on the Quality of English LanguageStrengths in Language Use
-
Professional Tone: The manuscript uses a formal academic tone appropriate for a scientific review.
-
Technical Terminology: Key terms such as "tumor-associated macrophages," "polarization," and "immune checkpoint inhibitors" are used correctly and consistently.
-
Logical Flow: The manuscript is structured logically, with sections that build on one another (e.g., TAM biology → immunoregulation → therapeutic targeting).
Areas for Improvement
-
Grammar and Sentence Structure:
-
Some sentences are overly long or complex, making them harder to follow. For example:
-
Original: "Macrophages, through their phagocytosis function, can initially clear tumor cells. However, when stimulated by TME factors, they eventually change into TAMs with the M2 phenotype, which increases metastasis and tumor growth by suppressing immunity, triggering angiogenesis, and bolstering cancer stem cells."
-
Suggested Revision: "Macrophages initially clear tumor cells through phagocytosis. However, exposure to factors in the tumor microenvironment (TME) induces their transformation into M2-polarized TAMs, which promote metastasis and tumor growth by suppressing immunity, inducing angiogenesis, and supporting cancer stem cells."
-
-
-
Wordiness:
-
Some phrases are unnecessarily verbose. For example:
-
Original: "It is anticipated that 2,041,910 new cases of cancer and 618,120 cancer-related deaths are projected to occur in the US in 2025."
-
Suggested Revision: "In 2025, 2,041,910 new cancer cases and 618,120 cancer-related deaths are projected in the US."
-
-
-
Repetition:
-
Certain ideas are repeated across sections (e.g., TAM polarization mechanisms). Consolidating these discussions would improve conciseness.
-
-
Clarity of Complex Concepts:
-
Some technical explanations could be simplified for better readability without losing scientific accuracy.
-
Example: "IL-4 binding to its receptor may phosphorylate STAT6, causing M2-like macrophage polarization via the JAK/STAT6 signaling pathway."
-
Suggested Revision: "IL-4 activates its receptor to phosphorylate STAT6, driving M2 macrophage polarization through the JAK/STAT6 pathway."
-
-
-
Transitions Between Sections:
-
Transitions between sections could be smoother to maintain a cohesive narrative. For example:
-
Between "Macrophage Polarization" and "TAMs in Immunoregulation," a brief linking sentence could be added: "Having discussed macrophage polarization, we now explore how these polarized TAMs influence immune regulation within the TME."
-
-
-
Use of Passive Voice:
-
The manuscript frequently uses passive voice, which can reduce clarity.
-
Example: "It has been shown that TAMs influence NK cell cytolytic activity in two ways."
-
Suggested Revision: "TAMs influence NK cell cytolytic activity through two mechanisms."
-
-
-
Typographical Errors:
-
Minor typographical errors should be corrected (e.g., inconsistent use of italics for gene names like STAT6).
-
-
Abstract Refinement:
-
The abstract is dense and could benefit from clearer structuring.
-
Original: "TAM's functions and mechanisms in PD-1/PD-L1 blocker resistance are described in detail."
-
Suggested Revision: "The manuscript details TAM functions and their role in resistance to PD-1/PD-L1 blockade."
-
-
Suggestions for Improvement
-
Simplify Complex Sentences:
Break down long sentences into shorter ones for improved readability. -
Avoid Redundancy:
Consolidate repeated discussions of TAM polarization pathways and roles. -
Enhance Transitions:
Add linking sentences between sections to improve narrative flow. -
Clarify Key Concepts:
Use simpler language to explain technical mechanisms while maintaining scientific rigor. -
Proofreading for Grammar and Style:
Conduct a thorough proofreading pass to correct minor grammatical errors and ensure consistency in terminology. -
Abstract Revision:
Rewrite the abstract to provide a concise summary of key findings without excessive detail.
Author Response
Point-by-point answer to the reviewer's 2 comments
Reviewer 2
Comments and Suggestions for Authors
Your work synthesizes current knowledge regarding the complex roles of TAMs in tumor progression, immunoregulation, and therapeutic applications. After careful evaluation, I offer the following assessment of your manuscript's strengths and areas for improvement.
Strengths
- Comprehensive Scope: Your review successfully covers multiple dimensions of TAM biology, from origins and polarization to complex interactions within the tumor microenvironment (TME). This breadth provides valuable context for readers across specialties.
- Clinical Relevance: The manuscript effectively bridges basic immunology with clinical applications, particularly in discussing immune checkpoint inhibitors and emerging immunotherapies. The connection to projected cancer statistics for 2025 establishes immediate relevance.
- Detailed Cellular Interactions: Your thorough examination of TAM interactions with various immune cells (NK cells, T cells, MDSCs, neutrophils, etc.) highlights the complexity of the TME and provides mechanistic insights into immunosuppression.
- Visual Representations: The figures appear well-conceptualized to illustrate complex biological processes, particularly the polarization pathways (Figure 1) and TAM-tumor cell interactions (Figure 2).
- Translational Focus: The emphasis on therapeutic targeting of TAMs (Section 7) demonstrates practical applications of the biological principles discussed earlier, enhancing the review's impact.
Areas for Improvement
Comment 1. Critical Analysis of Contradictory Evidence: While you present multiple pathways and mechanisms, the review would be strengthened by more robust discussion of conflicting evidence and unresolved controversies in the field. For example:
Comment. The dual roles of TAMs in pro/anti-tumor immunity
Answer: Thank you for the comment. We agree and have revised the manuscript accordingly. Please see Page 6, Line 217.
Comment. Contradictory results regarding TAM depletion versus reprogramming strategies
Answer: Thank you for the comment. We agree and have revised the manuscript accordingly. Please see Page 26, Line 878.
Comment. Inconsistent clinical outcomes from targeting CSF1R pathways
Answer: Thank you for the comment. We agree and have revised the manuscript accordingly for the clinical outcomes of CSF1R trials. Please see page 23, Lines 742 and 759.
Comment 2. Balanced Assessment of Clinical Efficacy: The conclusions regarding combination therapies (anti-PD-1 + TAM targeting) would benefit from more critical appraisal of clinical trial limitations, failures, and challenges in translating preclinical success to patient outcomes.
Answer: Thank you for the comment. We agree and have revised the manuscript accordingly for the clinical efficacy of anti-PD-1/PD-L1 Therapy. Please see page 20, Line 697.
Comment 3. Contemporary Context: While the foundation literature is well-represented, incorporating more recent breakthroughs (particularly single-cell RNA sequencing studies post-2020) would strengthen the review's currency and relevance.
Answer: Thank you for the comment. We agree and have revised the manuscript accordingly. We have added a section “2.2 Recent technological advances in TAM characterization”, with studies including single-cell RNA sequencing and related advances for TAM characterization. Please see page 3, Line 101.
Comment 4. Quantitative Synthesis: Consider incorporating meta-analytical approaches or quantitative synthesis of prognostic values across studies when discussing the clinical significance of TAM infiltration in various cancer types.
Answer: Thank you for the comment. We agree and have revised the manuscript accordingly. Please see page 24, Line 788.
Recommendations for Revision
- Expand discussion of contradictory findings and propose frameworks to reconcile these contradictions
Answer: Thank you for the comment. We agree and have revised the manuscript by expanding the discussion on contradictory findings. Please see page 26, Line 878.
- Provide more balanced assessment of clinical trial outcomes
Answer: Thank you for the comment. We agree and have revised the manuscript accordingly. Please see page 22, Line 718.
- Incorporate recent technological advances in TAM characterization
Answer: Thank you for the comment. We agree and have revised the manuscript accordingly. We have added a section “2.2 Recent technological advances in TAM characterization”, with studies including single-cell RNA sequencing and related advances for TAM characterization. Please see page 3, line 101.
- Consider adding a future directions section that proposes standardized approaches to TAM classification and biomarker discovery
Answer: Thank you for the comment. We agree and added the section in the revised manuscript. Please see page 27, Line 905.
Your manuscript provides valuable insights into TAM biology and therapeutic potential, addressing an important area in cancer immunology. With the suggested revisions, it would make an even stronger contribution to the field by enhancing methodological transparency and providing more nuanced analysis of complex, sometimes contradictory evidence.
Comments on the Quality of English Language
Strengths in Language Use
- Professional Tone: The manuscript uses a formal academic tone appropriate for a scientific review.
- Technical Terminology: Key terms such as "tumor-associated macrophages," "polarization," and "immune checkpoint inhibitors" are used correctly and consistently.
- Logical Flow: The manuscript is structured logically, with sections that build on one another (e.g., TAM biology → immunoregulation → therapeutic targeting).
Areas for Improvement
Comment 1. Grammar and Sentence Structure:
- Some sentences are overly long or complex, making them harder to follow. For example:
- Original: "Macrophages, through their phagocytosis function, can initially clear tumor cells. However, when stimulated by TME factors, they eventually change into TAMs with the M2 phenotype, which increases metastasis and tumor growth by suppressing immunity, triggering angiogenesis, and bolstering cancer stem cells."
- Suggested Revision: "Macrophages initially clear tumor cells through phagocytosis. However, exposure to factors in the tumor microenvironment (TME) induces their transformation into M2-polarized TAMs, which promote metastasis and tumor growth by suppressing immunity, inducing angiogenesis, and supporting cancer stem cells."
Answer: Thank you for the comment. We have revised the manuscript accordingly, breaking down the long sentences and carefully improving the language, as highlighted in red throughout.
Also, we have revised the statement mentioned above. Please see Line 57.
Comment 2. Wordiness:
- Some phrases are unnecessarily verbose. For example:
- Original: "It is anticipated that 2,041,910 new cases of cancer and 618,120 cancer-related deaths are projected to occur in the US in 2025."
- Suggested Revision: "In 2025, 2,041,910 new cancer cases and 618,120 cancer-related deaths are projected in the US."
Answer: Thank you for the comment. We have revised accordingly. Please see Line 24.
Comment 3. Repetition:
- Certain ideas are repeated across sections (e.g., TAM polarization mechanisms). Consolidating these discussions would improve conciseness.
Answer: Thank you for the comment. We have revised to remove the repetitions in the manuscript accordingly.
Comment 4. Clarity of Complex Concepts:
- Some technical explanations could be simplified for better readability without losing scientific accuracy.
- Example: "IL-4 binding to its receptor may phosphorylate STAT6, causing M2-like macrophage polarization via the JAK/STAT6 signaling pathway."
- Suggested Revision: "IL-4 activates its receptor to phosphorylate STAT6, driving M2 macrophage polarization through the JAK/STAT6 pathway."
Answer: Thank you for the comment. We have revised accordingly. Please see Line 84.
Comment 5. Transitions Between Sections:
- Transitions between sections could be smoother to maintain a cohesive narrative. For example:
- Between "Macrophage Polarization" and "TAMs in Immunoregulation," a brief linking sentence could be added: "Having discussed macrophage polarization, we now explore how these polarized TAMs influence immune regulation within the TME."
Answer: Thank you for the comment. We have revised and added transition sentences between all sections accordingly. Also, we have revised the statement mentioned above. Please see Line 362.
Comment 6. Use of Passive Voice:
- The manuscript frequently uses passive voice, which can reduce clarity.
- Example: "It has been shown that TAMs influence NK cell cytolytic activity in two ways."
- Suggested Revision: "TAMs influence NK cell cytolytic activity through two mechanisms."
Answer: Thank you for the comment. We have revised accordingly. Please see Line 372.
Comment 7. Typographical Errors:
- Minor typographical errors should be corrected (e.g., inconsistent use of italics for gene names like STAT6).
Answer: Thank you for the comment. We have corrected accordingly.
Comment 8. Abstract Refinement:
- The abstract is dense and could benefit from clearer structuring.
- Original: "TAM's functions and mechanisms in PD-1/PD-L1 blocker resistance are described in detail."
- Suggested Revision: "The manuscript details TAM functions and their role in resistance to PD-1/PD-L1 blockade."
Answer: Thank you for the comment. We have revised the abstract accordingly. Please see Line 14.
Suggestions for Improvement
- Simplify Complex Sentences:
Break down long sentences into shorter ones for improved readability.
Answer: Thank you for the comment. We have revised the manuscript accordingly to simplify complex sentences.
- Avoid Redundancy:
Consolidate repeated discussions of TAM polarization pathways and roles.
Answer: Thank you for the comment. We have revised the manuscript accordingly to avoid redundancy of TAM polarization pathways and roles.
- Enhance Transitions:
Add linking sentences between sections to improve narrative flow.
Answer: Thank you for the comment. We have revised the manuscript accordingly by adding transition sentences between sections.
- Clarify Key Concepts:
Use simpler language to explain technical mechanisms while maintaining scientific rigor.
Answer: Thank you for the comment. We have revised the manuscript accordingly.
- Proofreading for Grammar and Style:
Conduct a thorough proofreading pass to correct minor grammatical errors and ensure consistency in terminology.
Answer: Thank you for the comment. We have carefully revised the manuscript for grammar and style.
- Abstract Revision:
Rewrite the abstract to provide a concise summary of key findings without excessive detail.
Answer: Thank you for the comment. We have carefully revised the abstract with a concise summary.
We thank and truly appreciate the eminent reviewer's kind and comprehensive comments on our manuscript’s strengths and weaknesses.

Reviewer 3 Report
Comments and Suggestions for Authors
The authors presented a comprehensive review of macrophages in immuno-regulation. While the review is enriched with a lot of useful information, there are a few comments for the author to consider to improve the manuscript.
- Given the complex biology of TAMs, it would be interesting to talk about recent mathematical modeling work that integrates TAM physiology in cancer treatment and clinical trial simulation to facilitate translational research (e.g. PMID 35856032).
- There are indeed many reviews of TAMs. What is the new thing added by the current review?
Author Response
Point-by-point answer to the reviewer's 3 comments
Reviewer 3
Comments and Suggestions for Authors
The authors presented a comprehensive review of macrophages in immuno-regulation. While the review is enriched with a lot of useful information, there are a few comments for the author to consider to improve the manuscript.
Comment 1. Given the complex biology of TAMs, it would be interesting to talk about recent mathematical modeling work that integrates TAM physiology in cancer treatment and clinical trial simulation to facilitate translational research (e.g. PMID 35856032).
Answer: Thank you for the comment. We agree and have revised the manuscript accordingly. Please see page 24, Line 788.
Comment 2. There are indeed many reviews of TAMs. What is the new thing added by the current review?
Answer: Thank you for pointing this out. We agree there are many TAM reviews. But here, this review studies macrophage biology in TMEs, emphasizing their immunosuppressive and tumor formation functions and innovative treatment strategies based on recent literature. TAM cross-talk with immunoregulating immune cells was also precisely and thoroughly discussed. Next, it reviews TAM-targeted therapeutic contradictions, specifically between depletion and reprogramming, using TAM characterization advanced tools like single-cell sequencing and spatial transcriptomics. Unlike previous studies, it analyzes TAM’s bidirectional interaction with cancer cells to explain how they promote angiogenesis, metastasis, and immune evasion. Anti-PD-1/PD-L1 and macrophage-centric immunotherapy strategies are also discussed, along with the Integration of origin, polarization, and translational approaches. Finally, this article provides an elaborate framework for developing TAM-targeted precision cancer treatment based on recent advances.
We are very grateful for the insightful feedback the distinguished reviewer provided on our review.
